# Manipulating electron redistribution to achieve electronic pyroelectricity in molecular [FeCo] crystals

Pritam Sadhukhan[1,10], Shu-Qi Wu [1,10], Jeremy Ian Long[1], Takumi Nakanishi[1], Shinji Kanegawa[1✉], Kaige Gao[2], Kaoru Yamamoto[3], Hajime Okajima [4], Akira Sakamoto [4], Michael L. Baker [5,6], Thomas Kroll[7], Dimosthenis Sokaras[7], Atsushi Okazawa[8], Norimichi Kojima[9], Yoshihito Shiota [1], Kazunari Yoshizawa[1] & Osamu Sato [1✉]

Pyroelectricity plays a crucial role in modern sensors and energy conversion devices. However, obtaining materials with large and nearly constant pyroelectric coefficients over a wide temperature range for practical uses remains a formidable challenge. Attempting to discover a solution to this obstacle, we combined molecular design of labile electronic structure with the crystal engineering of the molecular orientation in lattice. This combination results in electronic pyroelectricity of purely molecular origin. Here, we report a polar crystal of an [FeCo] dinuclear complex exhibiting a peculiar pyroelectric behavior (a substantial sharp pyroelectric current peak and an unusual continuous pyroelectric current at higher temperatures) which is caused by a combination of Fe spin crossover (SCO) and electron transfer between the high-spin Fe ion and redox-active ligand, namely valence tautomerism (VT). As a result, temperature dependence of the pyroelectric behavior reported here is opposite from conventional ferroelectrics and originates from a transition between three distinct electronic structures. The obtained pyroelectric coefficient is comparable to that of polyvinylidene difluoride at room temperature.

[1] Institute for Materials Chemistry and Engineering & IRCCS, Kyushu University, 744 Motooka, Nishi-ku, Fukuoka, Japan. [2] College of Physical Science and Technology, Yangzhou University, Yangzhou, Jiangsu, People's Republic of China. [3] Department of Applied Physics, Okayama University of Science, Okayama, Japan. [4] Graduate School of Science and Engineering, Aoyama Gakuin University, 5-10-1 Fuchinobe, Chuo-ku, Sagamihara, Kanagawa, Japan. [5] The Department of Chemistry, The University of Manchester, Manchester, UK. [6] The Department of Chemistry, The University of Manchester at Harwell, Didcot, UK. [7] Stanford Synchrotron Radiation Lightsource, SLAC National Accelerator Laboratory, Stanford University, Menlo Park, CA, USA. [8] Division of Chemistry, Institution of Liberal Education, Nihon University School of Medicine, 30-1 Oyaguchi Kamimachi, Itabashi-ku, Tokyo, Japan. [9] Department of Basic Science, Graduation School of Arts and Sciences, The University of Tokyo, 3-8-1 Komaba, Meguro-ku, Tokyo, Japan. [10] These authors contributed equally: Pritam Sadhukhan, Shu-Qi Wu. ✉email: kanegawa@cm.kyushu-u.ac.jp; sato@cm.kyushu-u.ac.jp

Molecular materials based on dynamic electron processes are expected to provide better chemical tailorability, faster switching rates and light switchability[1–4]. The study of molecular compounds with superior physical properties attracts increasing attention because of their fundamental importance and potential applications in pyroelectric sensors[5–7], digital memory[8–11] and energy conversion devices[12–14]. One of the key properties is polarization[15–17]. In ferroelectric systems, the collective ion displacement and reorientation of polar molecules are normally utilized to induce changes in polarization[18, 19], and so is the intermolecular electron transfer in recent years[20–22]. Indeed, the emergent electron-based polarization properties can be realized by designing compounds whose electronic structure lies at the boundary of multiple states. A number of processes in dynamic molecular crystals, such as spin transition and intramolecular electron transfer, are accompanied by redistribution of electronic density[23]. Therefore, such compounds can be potentially used to induce polarization change and hence, the pyroelectric effect[24]. However, research in this regard is still scarce[25]. Pyroelectrics represent a class of non-centrosymmetric polar crystals which consists as inherent coupling between electrical polarization $P$ and temperature $T$, such that a temporal variation in temperature ($\delta T/\delta t$) results in a change in the electrical dipole moment, that is quantitatively described by the pyroelectric coefficient, $p = (\delta P/\delta T)$. Fundamentally, manifestation of the pyroelectric effect occurs as a temperature-dependent change in the surface-charge density which results in a pyroelectric current, $i_p = pA(\delta T/\delta t)$, where $A$ is the area of the measured surface. In conventional ferroelectrics, the polarization decreases upon heating towards the transition temperature and vanishes above this point[26–29]. However, in our molecular based approach, we can tune molecular dipole moment by changing the temperature regime. The change in polarization direction is predetermined by the electronic structures that dominate at different temperature regimes. Therefore, pyroelectric systems based on discrete molecules can induce pyroelectric properties when low-lying states with different dipole moments exist, and the gradual population change between such states upon temperature change occurs. To develop electronic pyroelectric compounds based on such considerations, we designed an [FeCo] dinuclear complex in which the ligand field strength around the $Fe^{3+}$ ion is in the spin crossover (SCO) region, and the redox-active deprotonated 2,5-dihydroxy-1,4-benzoquinone (dhbq) is used as a bridging ligand. Indeed, an unusual pyroelectric behavior, originating from the transition between three electronic states, is induced. Notably, the dhbq ligand allows for significant delocalization of electrons between the Fe 3d orbitals and the ligand $\pi^*$ orbitals. This delocalization leads to multiple electronic structure descriptions that can be better described as a quantum-mixing of configurations (as observed in some Ru-amine complexes or bio-inorganic systems like blue-copper protein or oxyhemoglobin[30–32]). For simplicity, we will use the limiting configurations, $[Fe^{3+}_{LS}–dhbq^{3-}–Co^{3+}_{LS}]$, $[Fe^{3+}_{HS}–dhbq^{3-}–Co^{3+}_{LS}]$ and $[Fe^{2+}_{HS}–dhbq^{2-}–Co^{3+}_{LS}]$, to denote their electronic structures and an extra superscript C (standing for covalent) is used to remind the readers of the presence of strong covalency in such states (Supplementary Note 1). At approximately 90 K, an abrupt spin transition from $Fe^{3+}_{LS}$ to $Fe^{3+}_{HS}$ occurs (LS = low spin, HS = high spin) which is followed by a temperature-induced continuous population change between the two redox isomers, $^C[Fe^{3+}_{HS}–dhbq^{3-}–Co^{3+}_{LS}]$ and $^C[Fe^{2+}_{HS}–dhbq^{2-}–Co^{3+}_{LS}]$, a phenomenon known as valence tautomerism (VT). Thus, pyroelectric effect emerges because of two distinct electronic dynamics, i.e. the change in spin and charge distribution in the polar [FeCo] crystal. A sharp pyroelectric current peak is observed at the spin transition temperature, whereas an approximately constant pyroelectric current is observed above the spin transition temperature

due to the successive population of the electron-redistributed $^C[Fe^{2+}_{HS}–dhbq^{2-}–Co^{3+}_{LS}]$ state. The crystal of the [FeCo] complex represents an intricate and ingenious example of electronic pyroelectricity, to which three electronic structures contribute.

## Results and discussion

**Synthesis and structural characterization.** Polar crystals of [FeCo] dinuclear complex were synthesized using the chiral ligands SS-cth and RR-cth (cth = 5,5,7,12,12,14-hexamethyl-1,4,8,11-tetra-azacyclotetradecane) as previously reported[33]. Enantiopure mononuclear complexes of Fe and Co, i.e., [Fe(AcO)(RR-cth)](PF$_6$) and [Co(AcO)(SS-cth)](PF$_6$), were mixed in a methanolic solution of dhbq in an equimolar ratio. The resulting mixture was then oxidized by AgPF$_6$, affording the complex [(Fe(RR-cth))(Co(SS-cth))(µ-dhbq)] (PF$_6$)$_3$ (**1(PF$_6$)$_3$**) as dark brown block-shaped crystals. The ESI-MS spectrum of **1(PF$_6$)$_3$** in acetone exhibits the molecular peak of the [FeCo] dinuclear complex ($m/z = 1111.5$) with a correct isotopic distribution, together with peaks attributable to [CoCo] and [FeFe] complexes (Supplementary Fig. 1). An elemental analysis including Co and Fe reveals that the experimental result is in agreement with the calculated one, where Co/Fe ratio is 1.00 (Supplementary Table 1). These results suggest that the selective crystallization of an [FeCo] heterometallic dinuclear complex occurs. The possible contamination from [FeFe][34] or [CoCo][35] species is well below the detection level of all the physical characterizations as discussed in later sections. A variable-temperature single-crystal analysis reveals that complex **1(PF$_6$)$_3$** crystallizes in the polar $P2_1$ space group (Supplementary Table 2). The corresponding structural data show that the discrete cationic molecules of $[1]^{3+}$ comprise $[\Lambda$-Fe(RR-cth)] and $[\Delta$-Co(SS-cth)] moieties bridged by a dhbq ligand (Fig. 1a). It is important to note that the Fe-to-Co orientation in each [Fe-dhbq-Co] unit is almost entirely aligned in the same direction along the b-axis in the structure of [(RR-cth)FeCo(SS-cth)]-[(RR-cth)FeCo(SS-cth)]-[(RR-cth)FeCo(SS-cth)] throughout the crystal, as restricted by the crystalline $2_1$-screw axis (Fig. 1b). As SCO in $Fe^{3+}$[36, 37] and valence tautomerism in Co[35, 38, 39] have been frequently observed in metal-catecholate systems, single-crystal structures obtained at various temperatures were compared[33, 36]. The bond lengths around the Fe ions (Fe–O, 1.904(5) Å; Fe–N, 2.026(5)–2.060(5) Å) and the Co ions (Co–O, 1.893(4) and 1.891(5) Å; Co–N, 1.987(5)–2.026(5) Å) at 60 K clearly reveal that both ions are in LS state (Supplementary

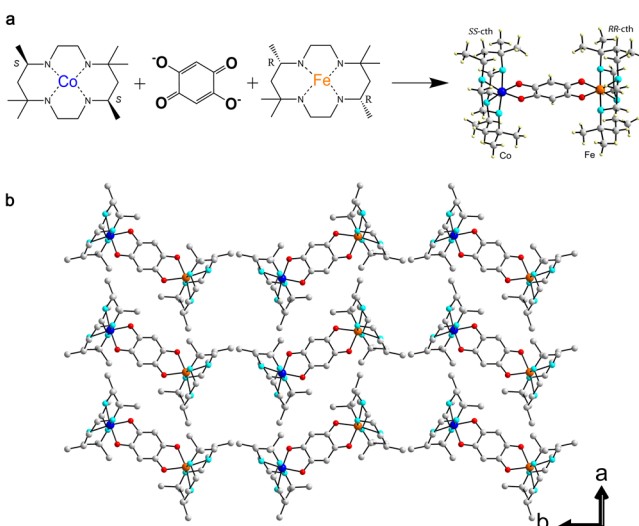

**Fig. 1 Synthetic strategy and crystal structure of the [FeCo] crystals.**
**a** Synthetic strategy toward the heterometallic dinuclear complex **1(PF$_6$)$_3$** [Fe(orange), Co(blue), O(red), C(gray)]; H atoms are omitted for clarity.
**b** Crystal packing of **1(PF$_6$)$_3$** at 130 K along the crystallographic c-axis.

Table 3, Supplementary Figs. 2 and 3). Upon heating to 100 K, the bond lengths around the Fe ions exhibit a significant elongation (Fe–O, 1.987(4) and 1.991(5) Å; Fe–N, 2.109(5)–2.145(5) Å), suggesting that the Fe ions are in HS state in this condition. The spin transition is driven by an entropy originating from the change in spin and vibrational contribution[40]. In contrast, the bond lengths around the Co ions remain almost unchanged (Co–O, 1.893(4) and 1.891(5) Å; Co–N, 1.979(5)–2.024(5) Å) and typical of the $Co^{3+}_{LS}$ state. The Fe-ligand bond distances were further compared at different temperatures which characterizes a gradual elongation from 100 K upto 300 K, representative of a progressive valence tautomerism process occurring over a wide temperature range[41] (Supplementary Fig. 2).

**Magnetic properties.** The magnetic properties of a polycrystalline sample of complex **1(PF₆)₃** were measured under an applied field of 0.1 T to determine the valence and spin state of the metal centers (Fig. 2)[42]. In the LT regime, the $\chi_m T$ product was determined to be 0.15 cm³ K mol⁻¹ at 5 K, indicating that the majority of the molecules populate a diamagnetic state ($S = 0$). On further heating, the $\chi_m T$ value abruptly increases at approximately 90 K, reaching to 3.00 cm³ K mol⁻¹ at 120 K. The ligand field around the Fe center is known to be appropriate for the spin crossover in $Fe^{3+}$ ion, therefore, the observed behavior can be inferred as the occurrence of the spin transition between the strongly antiferromagnetically (AF) coupled $Fe^{3+}_{LS}$–dhbq³⁻ and $Fe^{3+}_{HS}$–dhbq³⁻ states. A gradual increase up to 3.24 cm³ K mol⁻¹ is then observed upon heating to 300 K which corresponds to a system with a total spin quantum number ($S$) of 2. To be noted that we didn't find any change in $\chi_m T$ value in our data near the transition temperature (175 K) of the homometallic [CoCo] species which ensures the magnetic susceptibility data we obtained is genuinely originating from heterometallic [FeCo] contribution only[35].

**Pyroelectric properties.** The complex **1(PF₆)₃** crystallizes in the polar space group, enabling the observation of the pyroelectric effect from the dynamic change in the electronic structure upon varying the temperature. To unambiguously determine the direction of the pyroelectric current with respect to the single crystal, a plate-like piece of complex **1(PF₆)₃** crystal was indexed with silver paste attached to one side of the crystal surface and carbon paste attached to the parallel surface and the faces (010) and (0−10) were distinguished (Supplementary Fig. 4). The sign of the pyroelectric coefficient formally

corresponds to the electron transfer from dhbq to Fe in the whole temperature range upon heating. Below 60 K, only a weak current signal is detected upon heating, corresponding to a pyroelectric coefficient of less than 0.3 nC cm⁻² K⁻¹ (Fig. 3a). At a temperature approaching the abrupt transition point, the pyroelectric current exhibits a sharp increase, giving a peak as high as 30 nC cm⁻² K⁻¹, and then decreases to ca. 3.8 nC cm⁻² K⁻¹ after 120 K. This is clearly associated with both the structural change and the variation of electronic density due to the spin transition in the [FeCo] complex in a narrow temperature range. Integration over this temperature domain (80 to 120 K) gives a polarization change of approximately 0.48 μC cm⁻² (Fig. 3b). Upon further heating to room temperature, it is surprising to find that the pyroelectric current is still detected, and the pyroelectric coefficient remains at approximately 3.4 nC cm⁻² K⁻¹ at 300 K, which is as large as that of polyvinylidene difluoride (PVDF)[43]. We have also determined the pyroelectric response of the system by

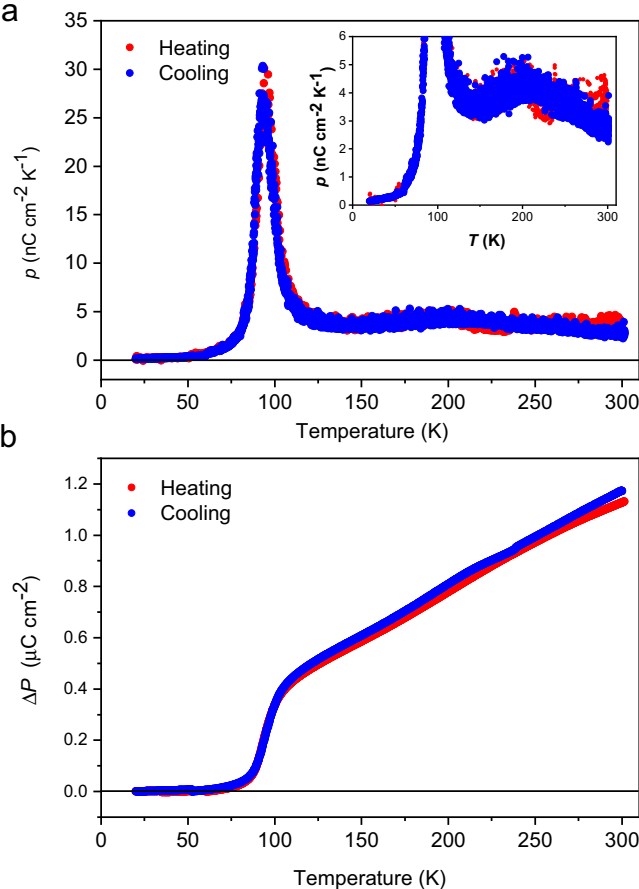

**Fig. 3 Pyroelectric properties of the [FeCo] crystals. a** Temperature dependence of the pyroelectric coefficient for **1(PF₆)₃**. Pyroelectric coefficients are almost constant at approximately 2.4−5.0 nC cm⁻² K⁻¹ between 300 and 120 K. Sharp peaks are obtained on both cooling and heating modes at ca. 90 K. These are consistent with the transition temperature observed in the magnetic measurements. No pyroelectric current is detected below the phase transition temperature. Data were collected with a temperature sweep rate of 5 K min⁻¹. Inset: expansion between 0−6 nC cm⁻² K⁻¹. **b** Temperature-dependent polarization change ($\Delta P$) plot with respect to the lowest temperature data (20 K) for **1(PF₆)₃**. The calculated polarization change is ca. 1.15 μC cm⁻² at 300 K. Data were collected on heating and cooling modes with a temperature sweep rate of 5 K min⁻¹.

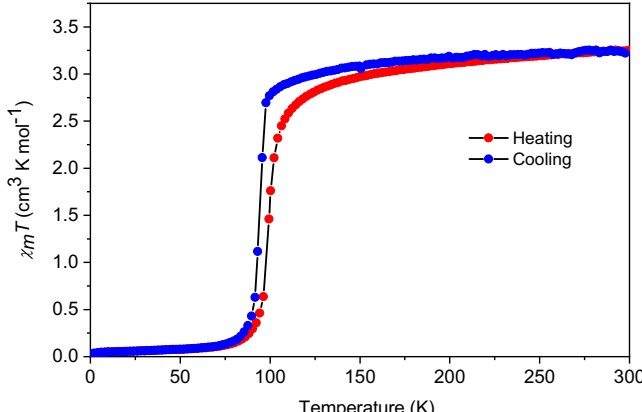

**Fig. 2 Magnetic properties of the [FeCo] crystals.** Temperature dependence of magnetic susceptibilities displayed as a plot of $\chi_m T$ vs. Temperature. Data were collected on heating and cooling modes with a temperature sweep rate of 2.5 K min⁻¹.

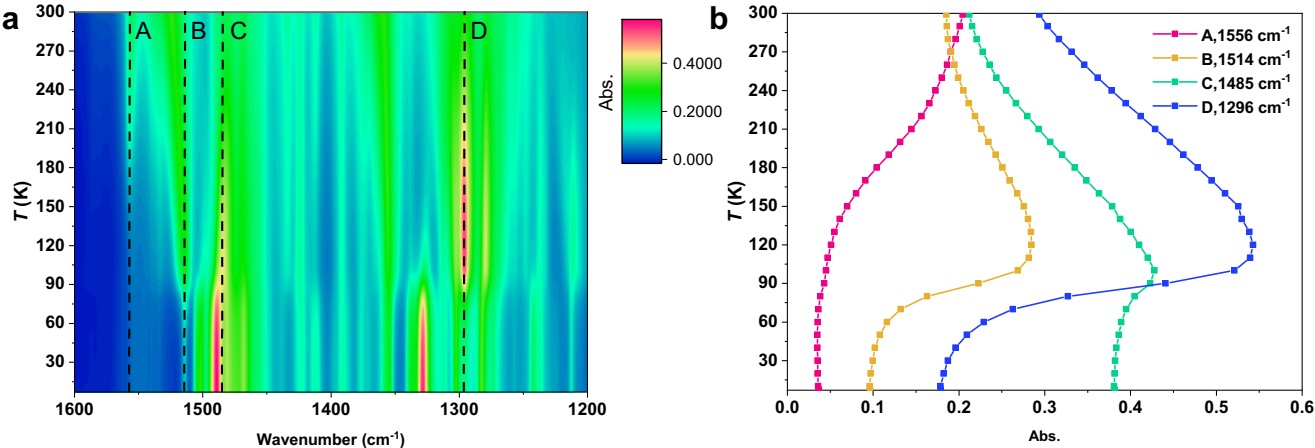

**Fig. 4 Variable-temperature IR spectroscopy of the [FeCo] crystals. a** 2D color map of the variable-temperature IR absorption spectra for the range of 1200–1600 cm$^{-1}$ for the temperature range of 300–10 K. Dashed lines represent the characteristic absorption bands. Abrupt change in absorbance at 1514 cm$^{-1}$ (line B) and 1296 cm$^{-1}$ (line D) is observed at approximately 90 K (Supplementary Fig. 8), which is consistent with the spin transition temperature. Even above spin transition temperature, a gradual change in absorbance is clearly detected. Absorbance at 1556 cm$^{-1}$ (line A) and 1485 cm$^{-1}$ (line C) gradually increases and decreases with increasing temperature, respectively. Absorbance at 1556 cm$^{-1}$ and 1485 cm$^{-1}$ is assignable to that of dhbq$^{2-}$ and dhbq$^{3-}$, respectively, suggesting that the increase of the population of $^{C}$[Fe$^{2+}_{HS}$–dhbq$^{2-}$–Co$^{3+}_{LS}$] state on the increment of temperature. **b** Temperature-dependent IR absorptions of selected wavenumbers shown in panel **a**.

continuous oscillation method (AC method)[44] from liquid nitrogen temperature to room temperature where the sample is subjected to thermal wave by a continuous, sinusoidally modulated laser of 0.1 Hz and the obtained data also shows continuous current release over a wide temperature range (Supplementary Fig. 5). In the continuous temperature ramping measurement, the charge trapped in lattice defects may be thermal excited and contaminate the pyroelectric signal. In this AC measurement, on the other hand, such trapped charges are swept out in the initial cycle of temperature modulation and their contribution to the time-averaged value is suppressed. Therefore, the method is believed to allow us to selectively detect the true pyroelectric current[45]. The appearance of the temperature-independent signal similar to the temperature ramping measurement strongly suggests that this signal component would be true pyroelectric current representing an electric polarization change. However, the pyroelectric property is known to be inherent to materials with polar structures. In the case of **1(PF$_6$)$_3$**, the pyroelectric behavior could stem not only from the possible intramolecular electron redistribution but also from a change in the strain of the crystal sample. To discriminate between these two contributions and to exclude the possibility that the pyroelectric current above the spin transition originates from a secondary pyroelectric effect due to the thermal deformation of the polar crystal, we measured the pyroelectric property of an isostructural [ZnCo] dinuclear complex (Supplementary Fig. 6 and Supplementary Table 5) as a reference material without temperature-dependent electron redistribution behavior. The pyroelectric coefficient of the [ZnCo] complex was found to be much smaller (< 15%) than that observed for **1(PF$_6$)$_3$** in the whole temperature range, supporting that the large pyroelectric coefficient of **1(PF$_6$)$_3$** mainly originates from a continuous electronic redistribution in the compound. Integration of the pyroelectric current as the temperature was changed from 120 to 300 K yielded a polarization change of approximately 0.67 μC cm$^{-2}$, which is significantly larger than that during the SCO process. Considering the relatively smaller change in Fe-ligand bond lengths observed in this temperature domain, the origin of the high-temperature pyroelectric effect should be assigned to a process

with more electron-transfer character rather than the incomplete SCO behavior.

**Infrared spectroscopy.** Further in-depth infrared spectroscopic investigation was carried out in pursuit of the origin of the unusual pyroelectric behavior in terms of the electronic structure determination at different temperature regime. At LT phase, infrared (IR) spectra were recorded from 7 K to 70 K with a 10 K interval but did not exhibit any significant change in the peak positions and intensities. The strong absorption band at 1485 cm$^{-1}$ is typical of the open-shell dhbq$^{3-}$ bridging ligand, as observed in similar [Co$^{3+}_{LS}$–dhbq$^{3-}$–Co$^{3+}_{LS}$] and [Cr$^{3+}$–dhbq$^{3-}$–Co$^{3+}_{LS}$] compounds[33, 39] supporting our assignment of the LT electronic structure of the $^{C}$[Fe$^{3+}_{LS}$–dhbq$^{3-}$–Co$^{3+}_{LS}$] state (Supplementary Fig. 7). IR measurements above 70 K evidence a change in the spectra at the transition temperature range (80–100 K), which consists of an abrupt decrease of the absorption bands around 1329 cm$^{-1}$ and the appearance of new bands at 1514 and 1296 cm$^{-1}$ (Supplementary Fig. 8). This means that the change in the spin state of the Fe ion is accompanied by variations in the electronic structure of the bridging ligand. Upon further increasing the temperature above 100 K, it was found that the peaks around 1485 cm$^{-1}$, characteristic of dhbq$^{3-}$, continuously decrease in intensity, whereas a new peak around 1556 cm$^{-1}$, characteristic of dhbq$^{2-}$, emerges and increases upon heating (Supplementary Fig. 8 and Fig. 4) as typically observed in reported valence tautomeric dinuclear compounds[33, 46]. Moreover, comparison of the variable temperature IR absorption spectra along with difference IR spectra of the [CrCo] dinuclear complex and the [FeCo] system suggested significant similarity between the high temperature behavior of [FeCo] and the charge-transfer phenomenon in [CrCo]; however, the low temperature behavior of [FeCo] complex shows the occurrence of completely different electron dynamics (Supplementary Figs. 9, 10 and 11)[33]. Such features reveal that in the HT phase there is a thermally accessible excited state, $^{C}$[Fe$^{2+}_{HS}$–dhbq$^{2-}$–Co$^{3+}_{LS}$], which gradually becomes the dominating phase in thermal equilibrium with the $^{C}$[Fe$^{3+}_{HS}$–dhbq$^{3-}$–Co$^{3+}_{LS}$] state. This is a signature of valence tautomerism that takes place over a large

span of temperature. These observations strongly support our assignment from pyroelectric measurements that the high temperature transition behavior comes from valence tautomerism whereas the low-temperature behavior is dominated by spin transition. The assignment is also consistent with the gradual increase of the $\chi_m T$ value in high temperature domain, which is caused by the gradual increase in the population of $Fe^{2+}_{HS}$ with unquenched orbital angular momentum, thus with an expected $\chi_m T$ large than that of strongly AF coupled $^C[Fe^{3+}_{HS}$–dhbq$^{3-}$–Co$^{3+}_{LS}]$ state.

**X-ray absorption and Mössbauer spectroscopy.** X-ray spectroscopic techniques are recently getting considerable attention in material science as it is efficient of probing the electronic as well as structural properties of various coordination complexes. X-ray absorption near edge spectroscopy (XANES) measurements at the Co K-edge support that Co sites remain unaltered during the transition process. Variable temperature measurements (at 9.5 K and 260 K) show only very subtle temperature dependence in the Co K-edge spectra, which indicates that the thermally induced spin transition must occur at the Fe site. The Co pre-edge includes one low intensity peak consistent with the 1 s → 3d quadrupole transition for octahedral $Co^{3+}_{LS}$. The $^1A_{1g}$ ground state has one allowed excited configuration, $t_{2g}^6 e_g^1$, resulting in a $^2E$ doublet excited absorption state that gives rise to observed single pre-edge feature (Fig. 5)[47].

Therefore, as the interplay between three distinct electronic states are presumed to be solely oriented around the iron center of the dinuclear moiety, Mössbauer spectroscopy supposed to be a very powerful technique to precisely determine the dynamic change of electronic configuration with temperature at Fe site. Hence, variable-temperature Mössbauer spectrum was recorded at 75 K, the isomer shift (IS) and quadrupole splitting (QS) of 0.42 and 1.64 mm s$^{-1}$, respectively, were obtained (Supplementary Fig. 12 and Table 4). The Mössbauer spectra of the [FeFe] species is absent in our Mössbauer spectra of [FeCo] species[34]. The obtained QS value is comparable to those observed in the LS state of catecholato-$Fe^{3+}$ SCO compounds decorated by electron-donating groups, which are well known to exhibit strong quantum mixing ground state hybridized from the $Fe^{3+}_{LS}$-catecholato and $Fe^{2+}_{LS}$-semiquinonato configurations[36, 48]. Fe K-edge XANES measurements performed at 8 K resolve a single low-intensity (5.1 units of intensity) pre-edge peak, which is typical for octahedral $Fe^{2+}_{LS}$ rather than $Fe^{3+}_{LS}$ (Fig. 6a)[49]. To investigate the pre-edge in greater detail, high energy resolution fluorescence detected (HERFD) XANES measurements were performed. The HERFD spectrum at 8 K resolves additional fine structure within the pre-edge region (Fig. 6b). Interpretation of these features matches neither $Fe^{2+}_{LS}$ or $Fe^{3+}_{LS}$ limiting descriptions. In the $Fe^{3+}_{LS}$ limiting description the HERFD spectrum includes a characteristic low-energy $^1A_{1g}$ ($t_{2g}^6 e_g^0$) absorption final state followed by a series of overlapping absorption final states ($^3T_{1g}$, $^3T_{2g}$, $^1T_{1g}$ and $^1T_{2g}$) at higher energy that originate from multiplet splitting within the $t_{2g}^5 e_g^1$ absorption configuration. In the $Fe^{2+}_{LS}$ limiting description the HERFD spectrum is dominated by just one absorption peak, relating to a $^2E_g$ ($t_{2g}^6 e_g^1$) absorption final state[49, 50]. Fitting the individual contributions to the 8 K HERFD spectrum of **1(PF$_6$)$_3$** requires a minimum of three peaks (Fig. 6b), supporting Mössbauer evidence that the ground state is composed of a strongly mixing combination of both $Fe^{2+}_{LS}$ and $Fe^{3+}_{LS}$. Therefore, we can identify the ground state of the [FeCo] complex as $^C[Fe^{3+}_{LS}$–dhbq$^{3-}$–Co$^{3+}_{LS}]$, or equivalently, the quantum-mixing state of the $[Fe^{3+}_{LS}$–dhbq$^{3-}$–Co$^{3+}_{LS}]$ and $[Fe^{2+}_{LS}$–dhbq$^{2-}$–Co$^{3+}_{LS}]$ configurations.

To understand the nature of the observed transition behavior and gain insight into the electronic structure at the HT phase, the Fe XANES measurements were performed at 250 K. The energy of the absorption edge threshold is consistent with $Fe^{3+}_{HS}$ ion (Fig. 5a). However, the relative intensities of the pre-edge peaks are inconsistent with quadrupole allowed transitions expected for octahedral $Fe^{3+}_{HS}$ ion (Fig. 6c). The lower energy peak is much less intense than the higher energy peak, which is inconsistent with the 3:2 peak ratio expected for $Fe^{3+}_{HS}$ with a $^5A_{1g}$ ($t_{2g}^3 e_g^2$) ground state and $^5T_2$ ($t_{2g}^4 e_g^2$) and $^5E$ ($t_{2g}^3 e_g^3$) final states. The pre-edge has 8.8 units of intensity, which fits within the range expected for distorted six coordinate $Fe^{2+}_{HS}$[51]. HERFD-XANES were also performed at 250 K which shows that the pre-edge structure is neither consistent with pure octahedral $Fe^{3+}_{HS}$ or $Fe^{2+}_{HS}$, providing further evidence of the increasing contribution from $^C[Fe^{2+}_{HS}$–dhbq$^{2-}$–Co$^{3+}_{LS}]$ configuration mixed in the electronic structure (Fig. 6d). These HERFD-XANES measurements provide almost identical pre-edge structure to those obtained at 135 K, and the origin of the high-temperature pyroelectricity remained unsettled (Supplementary Fig. 13). The Mössbauer spectrum recorded above the transition temperature (125 K) affords IS and QS values of 0.69 and 1.83 mm s$^{-1}$, respectively (Supplementary Table 4). While the increase in the IS value is consistent with spin transition behavior, the QS value is significantly larger than those observed in the HS state of catecholato-$Fe^{3+}$ SCO compounds[36] (typically, ~0.9 mm s$^{-1}$) and that of the $Fe^{3+}_{HS}$ state in reference $[Fe^{III}(cth)_2(dhbq)]^{3+}$ compound (~1.0 mm s$^{-1}$)[31]. Normally, the QS value decreases with pure spin transition from $Fe^{3+}_{LS}$ to $Fe^{3+}_{HS}$ state;[36] the observed increase in the QS value of **1(PF$_6$)$_3$** after the transition upon heating thus indicates that the electronic state just above the

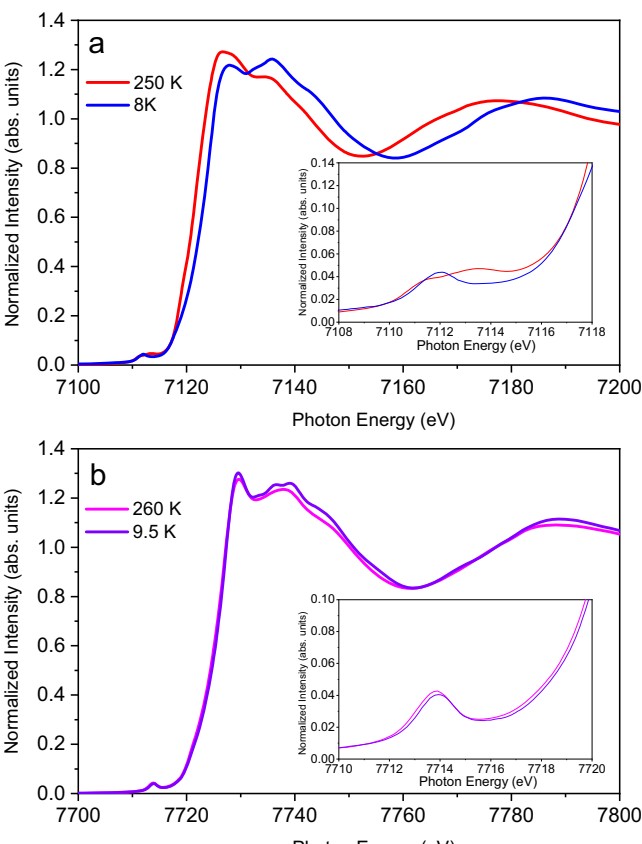

**Fig. 5 XAS of the [FeCo] crystals. a** Temperature-dependent Fe K-edge X-ray absorption spectroscopy (XAS) data for **1(PF$_6$)$_3$**. Inset- Fe pre-edge XAS data. **b** Co K-edge XAS data. Inset- Co pre-edge XAS data.

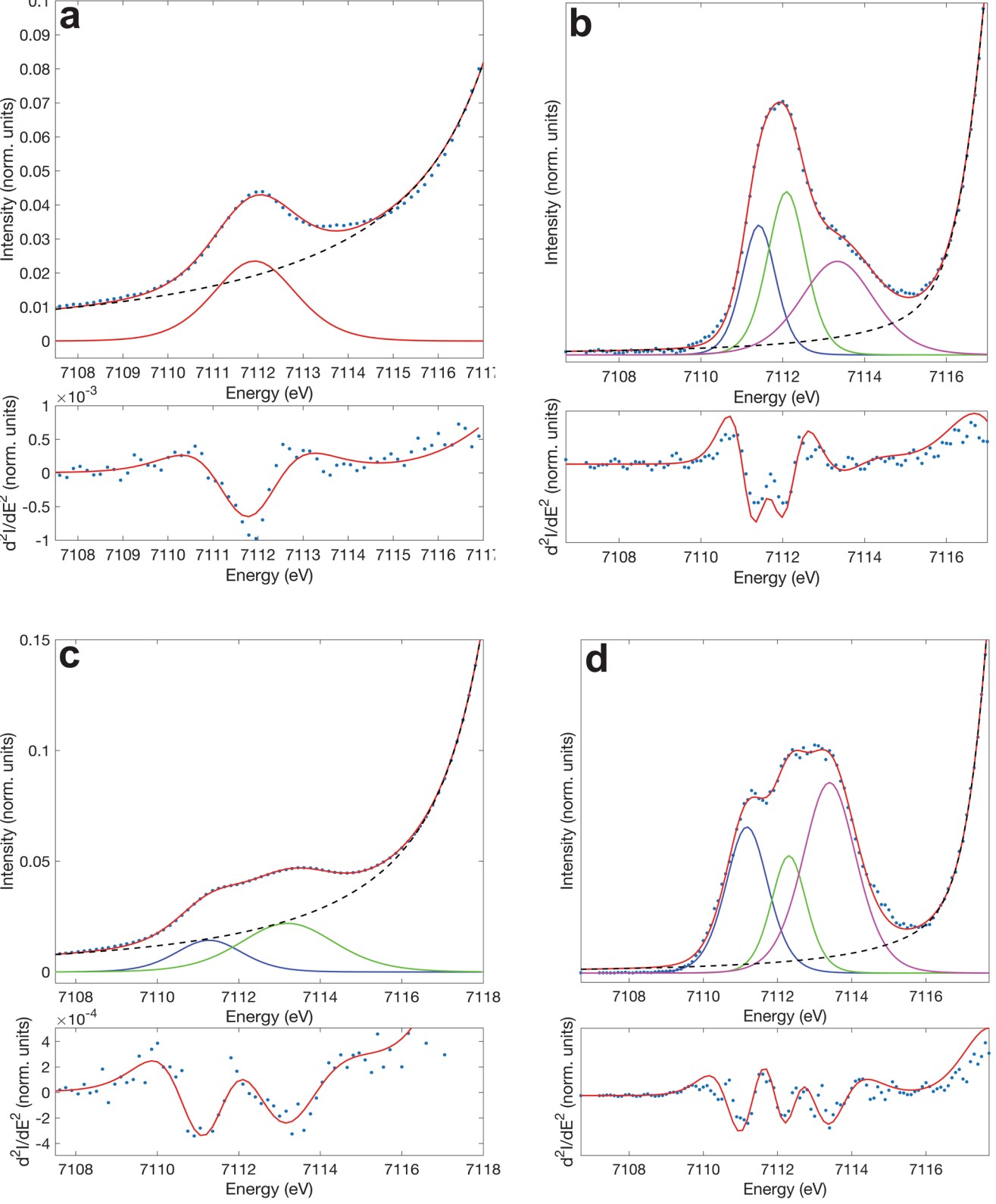

**Fig. 6 Fe-XANES and Fe HERFD-XANES recorded for the [FeCo] crystals. a** Upper figure: Fe-XANES pre-edge measured at 8 K (blue circles), fit to the rising edge (broken black line) and total fit (solid red line). Lower figure: second derivative of Fe-XANES measured at 8 K (blue circles) with related fit (solid red line). **b** Upper figure: Fe HERFD-XANES measured at 8 K (blue circles), fit to the rising edge (broken black line), individual peak fits (solid lines, blue, green, magenta) and total fit (solid red line). Lower figure: second derivative of Fe HERFD-XANES measured at 8 K (blue circles) with related fit (solid red line). **c** Upper figure: Fe-XANES pre-edge measured at 250 K (blue circles), fit to the rising edge (broken black line), individual peak fits (solid lines, blue, green) and total fit (solid red line). Lower figure: second derivative of Fe-XANES measured at 250 K (blue circles) with related fit (solid red line). **d** Upper figure: Fe HERFD-XANES measured at 250 K (blue circles), fit to the rising edge (broken black line), individual peak fits (solid lines, blue, green, magenta) and total fit (solid red line). Lower figure: second derivative of Fe HERFD-XANES measured at 250 K (blue circles) with related fit (solid red line).

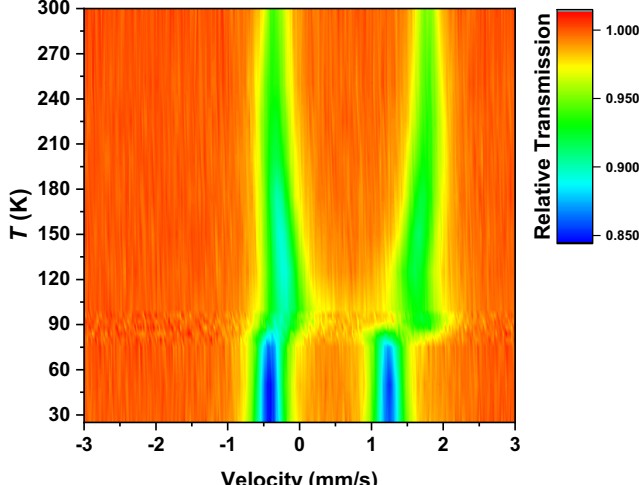

**Fig. 7 Variable-temperature Mössbauer spectroscopy of the [FeCo] crystals.** Temperature-dependent Mössbauer spectra of **1(PF₆)₃** between the temperature range from 300 to 25 K. Spin transition is observed at approximately 90 K. Upon heating above the spin transition temperature, the quadrupole splitting (QS) value is getting larger, which suggests that the contribution of the $^C[Fe^{2+}_{HS}–dhbq^{2-}–Co^{3+}_{LS}]$ state increases with temperature at the high-temperature phase.

transition temperature is not purely determined by the $Fe^{3+}_{HS}$ state. The $Fe^{2+}_{HS}$ state having a larger QS value (typically, 2.1–2.9 mm s$^{-1}$)[34] should also contribute via the population on the $^C[Fe^{2+}_{HS}–dhbq^{2-}–Co^{3+}_{LS}]$ state, which is consistent with the results from XANES measurements at the Fe K-edge.

The most interesting observation in the HT Mössbauer measurement is that up to higher temperatures (300 K), it shows a further gradual increase in the QS and IS values upon heating. However, no well-resolved peaks can be observed during this process, suggesting that the rates of electron hopping process are faster than the Mössbauer time window ($\tau \sim 10^{-8}$ s) but slower than the IR time window ($\tau \sim 10^{-12}$ s). It is well established that the quadrupole splitting in Mössbauer spectra usually decreases with pure spin transition with increasing temperature[36]. Therefore, the HT Mössbauer observation excludes the possibility of an incomplete spin transition and provides further evidence of the increasing population of the $Fe^{2+}_{HS}$ state with larger QS values at higher temperatures (Fig. 7). Combining the Mössbauer and IR spectroscopic studies reveals that the most plausible assignment for the HT electronic structure is that the $^C[Fe^{3+}_{HS}–dhbq^{3-}–Co^{3+}_{LS}]$ state dominates the system shortly after the SCO process, and then gradually evolves to the $^C[Fe^{2+}_{HS}–dhbq^{2-}–Co^{3+}_{LS}]$ state via VT process upon increasing temperature. This assignment is also consistent with the pyroelectric measurement that shows the HT process possesses more charge-transfer feature. Moreover, the reduction of net polarization change ($\sim$1.15 µC cm$^{-2}$) in the [FeCo] compound compared with the expected value for a typical VT transition ($\sim$2.05 µC cm$^{-2}$ for [CrCo] system) can be understood based on the quantum mixing model[33].

This study presented electronic pyroelectric crystals of the [FeCo] dinuclear complex [(Fe(*RR*-cth))(Co(*SS*-cth))(μ-dhbq)](PF₆)₃. The [FeCo] complex has three thermally accessible states, whose interconversion is induced upon changing the temperature. The ground state exhibits a $^C[Fe^{3+}_{LS}–dhbq^{3-}–Co^{3+}_{LS}]$ state and at approximately 90 K, it transforms into a $^C[Fe^{3+}_{HS}–dhbq^{3-}–Co^{3+}_{LS}]$ state due to spin transition. Further heating activates a process of valence tautomerism where the population of the $^C[Fe^{2+}_{HS}–dhbq^{2-}–Co^{3+}_{LS}]$ state increases with temperature, resulting in an almost continuous pyroelectric current comparable to that of PVDF

that is observed in a wide temperature range from 90 K up to room temperature. It should be noted that considering its time scale, electron paramagnetic resonance should be a very appropriate technique to investigate the electronic dynamics in our compound. Since the [FeCo] dinuclear complexes are designed to orient in the same direction within the crystal, the interconversion between the three states can be envisaged as the origin of the pyroelectric effect. Interestingly, these pyroelectric properties can be repeatedly observed in the absence of an electric field because the molecular orientation is fixed in the [FeCo] crystal, which contrasts with the typical ferroelectric behavior. Furthermore, the pyroelectric current is normally observed below the ferroelectric transition temperature in ferroelectrics. On the other hand, for the polar [FeCo] complex, the current is observed even at the HT phase. Thus, our work demonstrates the possibility that a large pyroelectric effect can be preserved up to high temperatures with a continuous current release. Note that while energy conversion from waste heat to electric power can be realized in high-temperature ferroelectrics, we show that dynamic molecular materials by sophisticated chemical design can be exploited for the same purpose without a ferroelectric transition.

## Methods

All solvents and reagents were used as received from Sigma-Aldrich company. Racemic-cth and the enantiopure ligands (*RR*-cth and *SS*-cth) (cth = 5,5,7,12,12,14-hexamethyl-1,4,8,11-tetraazacyclotetradecane) were prepared according to literature procedures[52]. All reactions were conducted under a dry N₂ atmosphere.

**Synthesis of enantiopure [Fe(AcO)(*RR*-cth)](PF₆).** A mixture of Fe(AcO)₂ (521 mg, 3.0 mmol) and *RR*-cth (850 mg, 3.0 mmol) in EtOH (10 mL) was heated to 60 °C under a N₂ atmosphere. After stirring at this temperature, solid NH₄PF₆ (540 mg, 3.3 mmol) was added to the solution. [Fe(AcO)(*RR*-cth)](PF₆) then gradually precipitated within an hour. After the reaction mixture was cooled in an ice-bath, a precipitate was collected by filtration and washed with cold EtOH followed by Et₂O. [Fe(AcO)(*RR*-cth)](PF₆) was obtained as an off-white crystalline solid (1.286 g, yield = 78%). Anal. C₁₈H₃₉N₄O₂F₆PFe (544.34) Calcd. C: 39.72, H: 7.22, N: 10.29; found C: 39.69, H: 7.21, N: 10.31.

**Synthesis of enantiopure [Co(AcO)(*SS*-cth)](PF₆).** A mixture of Co (AcO)₂·4H₂O (747 mg, 3.0 mmol) and *SS*-cth (850 mg, 3.0 mmol) in EtOH (10 mL) was heated to 60 °C under a N₂ atmosphere. After stirring at this temperature, solid NH₄PF₆ (540 mg, 3.3 mmol) was added to the solution. [Co(AcO)(*SS*-cth)](PF₆) then gradually precipitated within an hour. After the reaction mixture was cooled in an ice-bath, a precipitate was collected by filtration and washed with cold EtOH followed by Et₂O. [Co(AcO)(*SS*-cth)](PF₆) was obtained as light pink crystalline solid (1.23 g, yield = 75%). Anal. C₁₈H₃₉N₄O₂PCo (547.42) Calcd. C: 39.49, H: 7.18, N: 10.23; found C: 39.46, H: 7.19, N: 10.27.

**Synthesis of [(Fe(*RR*-cth))(Co(*SS*-cth))(μ-dhbq)](PF₆)₂.** A solution of 3,5-dihydroxy-1,4-benzoquinone (H₂dhbq; 140 mg, 1.0 mmol) and triethylamine (0.3 mL, 2.0 mmol) in MeOH (50 mL) was bubbled with dry N₂ gas for 10 min to remove oxygen. Subsequently, solid [Fe(AcO)(*RR*-cth)](PF₆) (544 mg, 1.0 mmol) and [Co(AcO)(*SS*-cth)](PF₆) (547 mg, 1.0 mmol) were added to the solution, and the reaction mixture was heated at 60 °C for 30 min. A hot solution of KPF₆ (276 mg, 1.5 mmol) in H₂O (30 mL) was added to this dark-red solution. The mixture was slowly cooled and then kept at room temperature. After two days, a brown precipitate of [(Fe(*RR*-cth))(Co(*SS*-cth))(μ-dhbq)](PF₆)₂ was collected by filtration and washed with H₂O and cold MeOH (630 mg, yield = 56%).

**Synthesis of [(Fe(*RR*-cth))(Co(*SS*-cth))(μ-dhbq)](PF₆)₃, 1(PF₆)₃.** Solid AgPF₆ (76 mg, 0.3 mmol) and H₂O (1 mL) were added to a solution of [(Fe(*RR*-cth))(Co (*SS*-cth))(μ-dhbq)](PF₆)₂ (335 mg, 0.3 mmol) dissolved in acetone (60 mL). After stirring for 10 min, the mixture was filtered to remove Ag. The filtrate was evaporated to dryness under reduced pressure, and crude **1(PF₆)₃** was collected as a brown solid and washed with a small amount of H₂O. The crude product was recrystallized from a mixed solvent of several drops of MeCN and hot H₂O/MeOH to afford dark-red plate-like single crystals suitable for structural analysis (274 mg, yield = 72.9%). Elemental analysis calcd. for C₃₈H₇₄N₈O₄F₁₈P₃FeCo: C: 36.2, H: 5.8, N: 8.65, Fe: 4.44, Co: 4.69; found: C: 36.41, H: 5.94, N: 8.92, Fe: 4.48, Co: 4.72; ESI MS (Supplementary Fig. 1): *m/z* = 1111.5 [**1(PF₆)₂**]⁺ (calcd for C₃₈H₇₄N₈O₄F₁₂P₂FeCo: 1111.75).

*X-ray Spectroscopy.* X-ray spectroscopy measurements were performed at the Stanford Synchrotron Radiation Lightsource (SSRL), operated at 3 GeV with an

electron beam current of 500 mA. The measured samples were maintained between 10 K and 270 K using an Oxford Instruments CF1208 continuous flow liquid helium cryostat. Orientation averaged powder samples were mixed in boron nitride to alleviate self-absorption effects. Monochromator energy calibration was performed using the first inflection point of an Fe foil set to 7111.2 eV and Co foil set to 7713.0 eV.

Fe K-edge X-ray absorption near edge structure (XANES) measurements were performed at beamline 9-3. The measurements were performed in transmission mode with $N_2$-filled ionization chambers and fluoresce measurements were collected at 90° to the incident beam using a total fluoresce diode with Soller slits and a Mn filter between the sample and the Soller slits. During measurement, the data in the Fe and Co K-edge and K pre-edge regions were continuously monitored in order to ensure sample integrity by comparing each individual scan to ones taken previously and minimal photodamage was observed. To minimize contributions from photodamaged product, different spots along the samples were scanned. The spin transition observed in variable temperature Fe K-edge XANES was confirmed to be reversible through a complete heating cooling cycle. Fe high energy resolution fluorescence detected (HERFD) X-ray spectroscopy measurements were performed at the undulator beamline 15-2. A double-crystal monochromator equipped with Si(311) crystals was used to select the energy with a resolution ($\Delta E/E$) of $\sim 10^{-5}$ and two Rh-coated Kirkpatrick-Baez mirrors set a 3.5 mrad delivered a 45-$\mu$m (v) x 800-$\mu$m (h) X-ray beam to the sample position. The (440) Bragg reflection of five Ge(110) crystals arranged in a 1 m Rowland geometry were used to select the emission energy with a single element silicon drift detector used to measure the X-ray emission[53]. The emitted beam path was enclosed by a He-filled bag to reduce the signal attenuation. A liquid He cooled cryostat for measurements at different temperatures was used. To avoid radiation damage, a sample stage that is equipped with motors to allow for horizontal and vertical movement for multiple sampling positions was employed. Radiation damage was carefully monitored through consecutive scans at the same spot and the exposure time/dose per irradiated spot was kept well below the levels required for noticeable beam-induced spectral alterations. For each HERFD spectrum the emission detector energy was set to the maximum of the Fe K$\alpha$1 emission line, 6404.52 eV for Fe$_{LS}$ and 6404.8 eV for Fe$_{HS}$.

Pre-edge peak fitting for both the XANES and HERFD results were performed using a Pearson VII line shapes with a fixed 50:50 ratio of Lorentzian to Gaussian functions where the energy positions, the full width half maximum and the peak heights were varied using a non-linear least squares fitting routine. The normalized XANES pre-edge intensity is defined as the total pre-edge peak area determined by trapezoidal numerical integration of the fitted peaks, multiplied by 100, where the post-edge of the XANES is normalized to unity.

**Pyroelectric measurements.** Pyroelectric measurements were performed via continuous temperature ramping technique with Keithley 6517B electrometer and a Quantum Design MPMS-XL chamber to control the temperature[54]. The single-crystal sample (surface area of 0.3036 mm$^2$) was sandwiched by silver and carbon paste on its (010) and (0-10) surfaces to determine the direction of the pyroelectric current (Supplementary Fig. 4). Data were also recorded on a crystal piece sandwiched by silver paste on both of its sides (Fig. 3) and the results are consistent. No electric field was applied to the sample. The measurement temperature was restricted between 10 and 300 K under a helium gas flow. The measurements were conducted at a temperature sweep rate of 5 K min$^{-1}$. Pyroelectric property by continuous oscillation mode[54] was performed in helium gas using a heat-exchange-gas-type cryostat. Temperature modulation was generated using a diode laser with a wavelength of 520 nm modulated at a frequency of 0.1 Hz. The power density at the sample surface was 1.6 mW mm$^{-2}$. The induced pyroelectric current was converted to the voltage signal with Stanford Research Systems SR570 current preamplifier and recorded with Signal Recovery 7256 dual-phase digital lock-in amplifier.

## Data availability

The supplementary crystal data can be obtained free of charge from the Cambridge Crystallographic Data Centre (www.ccdc.cam.ac.uk/data_request/cif) using identifiers CCDC 2025869-2025876. The authors declare that data supporting the findings of this study are available within the paper and its supplementary information files. Further data are available from the corresponding authors upon reasonable request.

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

## Acknowledgements
This work was supported by JSPS KAKENHI Grant Numbers JP20H00385, JP18K05057, JP21K05085, JP21K05086, JP20K05421, JST-CREST JPMJCR15P5, JST-Mirai JPMJMI18A2 and the MEXT Project of "Integrated Research Consortium on Chemical Sciences". The synchrotron radiation experiments were performed at the BL02B1 of SPring-8 with the approval of the Japan Synchrotron Radiation Research Institute (JASRI) (Proposal No. 2019B1272, 2020A1124, 2021A1070). Use of the Stanford Synchrotron Radiation Lightsource, SLAC National Accelerator Laboratory, is supported by the U.S. Department of Energy, Office of Science, Office of Basic Energy Sciences under Contract No. DE-AC02-76SF00515. A part of this work was performed with the aid of Instrument Center, Inst. Mol. Sci. Okazaki.

## Author contributions
O.S. and S.K. supervised the project. P.S., J.I.L. and T.N. carried out synthetic and crystallographic experiments. Pyroelectric measurement was performed by K.G., J.I.L., S.K. and K.Y. M.L.B. measured the XANES spectra. M.L.B., T.K. and D.S. measured and analyzed the HERFD spectra. A.O. and N.K. undertook the spectra of Mössbauer. H.O. and A.S. performed the IR measurements. S.-Q.W., Y.S. and K.Y. have carried out theoretical analysis. S.-Q.W., P.S., S.K. and O.S. discussed and co-wrote the manuscript.

## Competing interests
The authors declare no competing interests.
