## [Peer Review File · Nature Communications]

REVIEWER COMMENTS

Reviewer #1 (Remarks to the Author):

This manuscript by Sato, Kanegawa and coworkers report on the synthesis, structural, magnetic and pyroelectric characterization of an enantiomerically pure Iron-cobalt tetraoxolene complex which undergoes a spin crossover (SCO) transition around 90 K. The authors report a pyroelectric current which reaches a peak at the SCO transition but then remains roughly constant at relatively high value, comparable to that of polyvinylidene difluoride up to room temperature. This is of particular relevance for potential applications. A detailed spectroscopic characterization leads them to suggest that this peculiar behavior is due to the contributions of three different electronic states.

The reported results can be of interest for the broad readership of Nature Communications, showing how a combination of labile electronic structures with appropriate crystal engineering may lead to pyroelectricity of purely molecular origin. The novelty of the result is however partially reduced by two previous reports of the same group: one concerning pyroelectricity observed in a Co-dioxolene Valence tautomeric system (ref. 25); the second one concerning the previously reported synthesis, structure and characterization of the isostructural CrCo complex (ref. 30). Nonetheless, I think that the reported properties are interesting enough to deserve, in principle, publication in Nature Communications. There are, however, some major issues to be solved before acceptance can be granted. These are detailed in the following:

- 1- First, and foremost, the manuscript is quite badly organized and unclearly written in different sections. The section Discussion actually contains most of the results reported in the manuscript; at the same time, the Results section already anticipate some Discussion. In this respect, a single Results and Discussion section would be preferable; otherwise, a more accurate distinction between facts (Results) and interpretation (Discussion) is required. At the same time, I find the title quite unclear: what is a "dynamic" crystal? The described effects are temperature dependent, but this does not show up in the title. The studied system is a molecule, and this also is not clear from the title.
- 2- One of the main issues when dealing with heterometallic M-M' systems is obviously that of avoiding metal scrambling, i.e. the obtainment of M-M and M'-M' complexes together with the sought after M-M' couple. In the synthetic part authors describe ESI-MS spectra which clearly points to the obtainment of these unwanted complexes, but then conclude – without any other proof at this stage - that only the desired FeCo is obtained. While I am convinced of this result after having read the whole manuscript, this only becomes clear after magnetic and XAS discussion. Structural data by themselves are not enough (due to very small difference between Fe and Co), and the reference to 30 can be misleading, since there no evidence of the homonuclear couples was reported. This has to be changed.
- 3- Authors provide an explanation of the magnetic, spectroscopic and pyroelectric data which involve the contribution of a Fe(II)HS-DHBQ(2-) state close-lying (and increasingly populated on increasing temperature) to the Fe(III)HS-DHBQ(3-), with strong AF coupling. I am quite surprised to see that the latter "could not be obtained by DFT calculations". This casts several doubts as to the appropriateness of such hypothesis. As a matter of fact, the DFT results are presented in a very scattered and unclear way along the manuscript, especially for what concerns the energies of the different configurations, whereas they would be of paramount importance. In this respect, any future version of the manuscript should include much more details on the results obtained by DFT, possibly in tabular material in the ESI.
- 4- Directly linked to point 3 is the consideration that the observed room temperature value of χT is more in agreement with an $S=2$ with g slightly larger than 2 (as expected for Fe(II)) than for an $S=2$ arising from Fe(III) and radical species, for which $g=2$ is expected (in this case, the value of 3.12 and the decrease on decreasing temperature might suggest an incomplete AF coupling at this temperature). Here again DFT calculation could shed light on the expected Fe(III)-radical coupling.
- 5- An important point also concerns the definition of the supposed shift from Fe(III)HS-DHBQ(3-) species to Fe(II)HS-DHBQ(2-) configuration at high temperature: is there any peculiar reason why the

authors refrain to identify this as a Valence tautomerism (intramolecular redox isomerism)? If there is a difference, this should be made explicit; I suppose that simple quantum mixing, such as the one described in ref. 39, should be temperature independent. I further notice that this electron hopping occurs, according to authors, at a rate faster than the Mössbauer time window ($\tau \sim 10^{-8}$ s) but slower than the IR time window ($\tau \sim 10^{-12}$ s). In this respect, it looks like EPR should be a very appropriate technique to monitor this process: author should consider to apply it to this system and, at any rate, comment on this possibility.

6- Last, but not least: I found it quite astonishing not to find referenced the paper of Prof. A. Dei on the original dinuclear Co system (ACIE 2004) featuring CTH as ancillary ligand, which is far more appropriate than the current ref. 33, where TPA derivatives were used.

Reviewer #2 (Remarks to the Author):

The manuscript entitled "Manipulating electron redistribution to achieve exotic electronic pyroelectricity in dynamic [FeCo] crystals" by Prof. Shinji Kanegawa and Prof. Osamu Sato describes the pyroelectricity of single crystal of a bimetallic FeCo coordination compound crystallising in a polar space group. As the authors point out, the use of iso-oriented intramolecular electronic transition in molecular crystal is a promising new method to prepare materials with switchable electronic polarization. To support their conclusions, the authors present a wide temperature dependent characterisation of the electronic states involved in the transition (temperature dependence of X-ray diffraction data, X-Ray Absorption Spectroscopy at the Fe and Co K-edges, Mossbauer and infrared spectroscopies, magnetometry and pyroelectricity).

The approach to the development of a new material featuring electronic pyroelectricity based on switchable polarization is very elegant, since it involves a crystal engineering technique to crystallise a complex showing valence tautomerism in a polar space group. Anyway, I believe that the present manuscript does not fulfil the requirements to be published in the Nature Communications journal in terms of novelty of the results. First of all, a similar approach has been already reported previously by the same authors (reffer. 25 and 30 in the text). In ref. 30, in fact, the authors used the same enantiopure CTH ligands to crystallise the corresponding CrCo dinuclear complex in a polar space group, while in reference 25 the same authors reported the pyroelectricity associated with a valence tautomeric transition of a cobalt complex for the first time. These findings, in my opinion, affect the novelty of the presented results in a significant manner. Moreover, the novelty consisting in the continuous current release reported in the present work is somehow reduced by the fact that the pyroelectric coefficient of the compound $[\text{Co}(\text{phendiox})(\text{rac-cth})](\text{ClO}_4) \cdot 0.5\text{EtOH}$, reported in reference 25, stayed above the 5.0 nC/Kcm^{-2} limit for a wide temperature range as well (200 – 330 K). With these considerations in mind, I suggest publishing the present manuscript in a more specialised journal.

Minor typos:

- Page 3: Replace "Fundamentally, manifestation of the pyroelectric effect occurs as a temperature-dependent change in the surface-charge density results" with "Fundamentally, manifestation of the pyroelectric effect occurs as a temperature-dependent change in the surface-charge density which results"
- Page 9: replace " $[\text{Fe}(\text{cth})_2(\text{dmbq})]^{3+}$ " with " $[\text{Fe}_2(\text{cth})_2(\text{dmbq})]^{3+}$ "
- Page 11: replace "between mixed-valence electronic states" with "between mixed-valence electronic states"

Reviewer #3 (Remarks to the Author):

In this manuscript, the authors synthesized a novel polar crystal $[(\text{Fe}(\text{RR-cth}))(\text{Co}(\text{SS-cth}))(\mu\text{-dhbq})](\text{PF}_6)_3$ and reported its pyroelectric effect. It is interesting that different from the conventional pyroelectric effect in ferroelectrics, the polarization of the molecular material $[\text{FeCo}]$ dinuclear complex originates from a combination of Fe spin transition and redistribution of electron density between redox isomers of high-spin Fe through a charge transfer between the Fe atom and redox active ligand, which can be changed by the change of temperature, yielding the pyroelectric response. Even though the pyroelectric coefficient is inferior to those of conventional ferroelectrics including ceramics and single crystals, the sizable pyroelectric coefficient can be observed in a wide temperature range over a ~ 200 °C temperature span. The basic idea and hypothesis of the results are supported by applying several experimental data and DFT calculations. It is believed that this work would motivate further explorations of pyroelectric materials, especially pyroelectric features in molecular materials. Though the work is interesting, there are still several concerns that are suggested to be addressed.

1. The authors claimed that "To unambiguously determine the direction of the pyroelectric current with respect to the single crystal, a plate-like piece of complex $1(\text{PF}_6)_3$ crystal was indexed with silver paste attached to one side of the crystal surface and carbon paste attached to the parallel surface and the faces (010) and (0-10) were distinguished" in the section of Pyroelectric property. How did the authors determine the crystal faces? Please provide credible characterization. Moreover, how did the authors get the plate-like piece of the complex crystal? A photo of single-crystal should be given, in which the crystal faces should be marked.

2. As shown in Fig. 4b, the pyroelectric current could be detected even over the transition temperature. It is well-known that the crystal defects seriously affect the pyroelectric current with the increasing temperature. Therefore, the measurements of dielectric constant/loss and resistivity versus temperature should be supplied.

3. It is believed that the statement "the conventional pyroelectric effect in ferroelectric systems suffers from a significant temperature-dependent decrease upon cooling from the transition temperature, severely restricting their working temperature ranges" is inappropriate. In conventional ferroelectrics, for example, lead zirconate titanate (PZT) and lead magnesium niobite- lead titanate (PMN-PT), much larger pyroelectric coefficients can be obtained in a wide temperature range. The commercial PIR detectors equipped with PZT ceramics can work in both winter and summer regardless of regions from the equator to the high latitude area.

4. According to the demonstration in the manuscript, it is found that the polarization property of $[(\text{Fe}(\text{RR-cth}))(\text{Co}(\text{SS-cth}))(\mu\text{-dhbq})](\text{PF}_6)_3$ is quite different from that of the traditional pyroelectric materials. The polarization of $[(\text{Fe}(\text{RR-cth}))(\text{Co}(\text{SS-cth}))(\mu\text{-dhbq})](\text{PF}_6)_3$ crystal increases with the increase of temperature, which, however, decreases in ferroelectric materials (except part of antiferroelectric). Why does the polarization increase with the temperature in the molecular material? The nature of the positive pyroelectric coefficient is suggested to be discussed in detail.

5. More experimental details should be given:

1) The information of the reagents used in the experiment should be added.

2) The silver and carbon pastes were used as the electrodes in the experiment. How did the authors control the sizes of the electrodes? And what is the size of the electrode?

6. The authors should pay more attention to the format of the units:

1) There are three kinds of unit formats in this manuscript: "5 K·min⁻¹", "5 K/min", and "1.15 $\mu\text{C cm}^{-2}$ ".

2) In the pictures, what should be used to distinguish the physical quantities and corresponding units, "/" or "()" ?

3) "ml" should be "mL". Please check the mistakes like this in the whole manuscript.

Response to Reviewers

Review #1

This manuscript by Sato, Kanegawa and coworkers report on the synthesis, structural, magnetic and pyroelectric characterization of an enantiomerically pure Iron-cobalt tetraoxolene complex which undergoes a spin crossover (SCO) transition around 90 K. The authors report a pyroelectric current which reaches a peak at the SCO transition but then remains roughly constant at relatively high value, comparable to that of polyvinylidene difluoride up to room temperature. This is of particular relevance for potential applications. A detailed spectroscopic characterization leads them to suggest that this peculiar behavior is due to the contributions of three different electronic states. The reported results can be of interest for the broad readership of Nature Communications, showing how a combination of labile electronic structures with appropriate crystal engineering may lead to pyroelectricity of purely molecular origin. The novelty of the result is however partially reduced by two previous reports of the same group: one concerning pyroelectricity observed in a Co-dioxolene Valence tautomeric system (ref. 25); the second one concerning the previously reported synthesis, structure and characterization of the isostructural CrCo complex (ref. 30). Nonetheless, I think that the reported properties are interesting enough to deserve, in principle, publication in Nature Communications. There are, however, some major issues to be solved before acceptance can be granted. These are detailed in the following:

Thank you very much for reviewing our manuscript. We believe your valuable suggestions would add more value to our work and increase the quality of this manuscript. We have tried to answer your concerns and have done suitable corrections according to your suggestions.

Point 1. First, and foremost, the manuscript is quite badly organized and unclearly written in different sections. The section Discussion actually contains most of the results reported in the manuscript; at the same time, the Results section already anticipate some Discussion. In this respect, a single Results and Discussion section would be preferable; otherwise, a more accurate distinction between facts (Results) and interpretation (Discussion) is required. At the same time, I find the title quite unclear: what is a "dynamic" crystal? The described effects are temperature dependent, but this does not show up in the title. The studied system is a molecule, and this also is not clear from the title.

As the Reviewer suggested, Results section includes some discussion and Discussion section contains a lot of experimental results, which might be a bit misleading to the readers. According to the reviewer's comments, we have reconstructed the manuscript. In the current version, under the 'Results' section, we have serially introduced synthesis and structural characterization,

magnetic property and pyroelectric property one by one, followed by the new subheading added ‘spectroscopic and theoretical studies’, where we demonstrated Infra-red, Mossbauer, XAS and theoretical calculations. Results are presented with respect to different temperature domains (LT and HT phenomenon). We hope you will find our rearrangement satisfactory.

Furthermore, we have deleted the term, dynamic, from the title of the manuscript and added the term, molecule. The term, temperature dependent, is not included, because pyroelectric current is based on temperature change. New title is ‘Manipulating electron redistribution to achieve exotic electronic pyroelectricity in molecular [FeCo] crystals’. Thank you for the suggestions.

Point 2. One of the main issues when dealing with heterometallic M-M’ systems is obviously that of avoiding metal scrambling, i.e. the obtainment of M-M and M’-M’ complexes together with the sought after M-M’ couple. In the synthetic part authors describe ESI-MS spectra which clearly points to the obtainment of these unwanted complexes, but then conclude – without any other proof at this stage - that only the desired FeCo is obtained. While I am convinced of this result after having read the whole manuscript, this only becomes clear after magnetic and XAS discussion. Structural data by themselves are not enough (due to very small difference between Fe and Co), and the reference to 30 can be misleading, since there no evidence of the homonuclear couples was reported. This has to be changed.

Thank you for figuring out this point. Although, in Supp. Fig. 1, we have discussed about the possibility of metal scrambling in solution phase, we should make it clearer in the main text. Indeed, we confirmed the selective crystallization of the [FeCo] species after the thorough comparison among the solid-state spectroscopic data of [FeCo] and already reported homometallic [FeFe] and [CoCo] species [**Ref. 31**- Dei, A., Gatteschi, D., Pardi, L. & Russo, U. Tetraoxolene radical stabilization by the interaction with transition-metal ions. *Inorganic Chemistry* **30**, 2589-2594, doi:10.1021/ic00012a006 (1991); **Ref. 32**- Carbonera, C., Dei, A., Létard, J.-F., Sangregorio, C. & Sorace, L. Thermally and Light-Induced Valence Tautomeric Transition in a Dinuclear Cobalt–Tetraoxolene Complex. *Angewandte Chemie International Edition* **43**, 3136-3138, doi:10.1002/anie.200453944 (2004).]. Mossbauer spectrum of the [FeFe] species is absent in our Mossbauer spectra of [FeCo] complex. Moreover, Ref. 32 describes the inherent magnetic properties of the valence tautomeric [CoCo] complex which undergoes a valence tautomeric transition at 175 K; while in comparison with the [FeCo] magnetic properties, we didn’t find any change in χT value in our data near 175 K. This comparative study clearly

signifies that even if trace amount of [FeFe] or [CoCo] species is generated, quantitatively it is below detection level of the magnetic or Mossbauer spectroscopic techniques. Besides, the novel pyroelectric behavior is obtained while the measurement is pursued over a randomly selected single crystal of the [FeCo] species. Therefore, our results and conclusion remain same and would not be affected by the homometallic impurity or metal scrambling in solution phase. To make our point clearer, we have included explanations mentioned above in the corresponding sections, and have added one sentence, “the possible contamination from [FeFe] and [CoCo] species is well below the detection level of all the physical characterizations as discussed in later sections”, after mentioning the selective crystallization of the [FeCo] in the Synthesis section.

We have removed ref. 30 and cited the synthesis of homonuclear couples instead for clarity.

We hope our explanation and revision will be satisfactory.

Point 3. Authors provide an explanation of the magnetic, spectroscopic and pyroelectric data which involve the contribution of a Fe(II)HS-DHBQ(2-) state close-lying (and increasingly populated on increasing temperature) to the Fe(III)HS-DHBQ(3-), with strong AF coupling. I am quite surprised to see that the latter “could not be obtained by DFT calculations”. This casts several doubts as to the appropriateness of such hypothesis. As a matter of fact, the DFT results are presented in a very scattered and unclear way along the manuscript, especially for what concerns the energies of the different configurations, whereas they would be of paramount importance. In this respect, any future version of the manuscript should include much more details on the results obtained by DFT, possibly in tabular material in the ESI.

Thank you for your suggestions. We have included the schematic presentation of our DFT results corresponding to the computed electronic configurations of HS and LS in Supplementary Fig. 8. However, as we indicated in the manuscript, the AF Fe(III)HS-DHBQ(3-) state cannot be obtained by the current state-of-the-art computational methods. The convergence failure is partly due to the non-dynamical correlation error for strongly correlated systems. Besides, it can also be found that the Fe(II)HS-DHBQ(2-) state is the ground-state on the spin manifold, which is also unreasonable for a compound that exhibits thermal transition from AF Fe(III)LS-DHBQ(3-) to Fe(II)HS-DHBQ(2-) state (considering the latter state is entropy-favored). This should originate from the single-determinant formalism (such as DFT) cannot treat the non-dynamical correlation adequately in this compound. Therefore, we didn't assign the electronic structures based on the relative energies of different electronic states, but determine them by comparing the

IR absorption features to the computed vibrational frequencies. Considering these, the DFT calculation section was arranged together with the IR spectra in the manuscript. Furthermore, we have performed more sophisticated post-Hartree-Fock calculations with the complete active space self-consistent field (CASSCF) to allow the mixing of various electronic configurations. The active space was chosen as the minimal one containing all the ten metal 3d-centered orbitals and one metal-dhbq π bonding orbital. The dynamic correlation was included by using the N-electron valence state perturbation theory (NEVPT2) based on the crystalline geometry at 100 K. Indeed, the strongly covalent Fe(III)HS-DHBQ(3-) state can be identified as the ground state. However, the chemical scenario of its electronic structure is even more complicated. The calculated low-lying Fe $d\pi$ - π bonding orbital has greater metal contribution of ca. 54.6% than that of its antibonding partner (38.4%). This feature lends certain Fe(II)HS character to the Fe center, leading to the quantum mixing of states as discussed in the manuscript. The ground state, on the other hand, is represented as the hybridization of three main components: both electrons on $d\pi$ - π bonding orbital (35.1%), one electron on $d\pi$ - π bonding orbital and the other one on $d\pi$ - π antibonding orbital (36.7%) and both electrons on $d\pi$ - π antibonding orbital (23.1%). It is a highly complicated case. However, even the minimal active space was chosen, it was still extremely time consuming (about 1 month) to converge the wavefunction, which limits further improvements with current computational technology. Furthermore, it is a formidable task to calculate the IR spectra at such a high level of calculation. Therefore, it is unfortunately not possible to apply CASSCF NEVPT2 analysis to further our understanding of **1**(PF₆)₃. We hope that you will find our explanation satisfactory.

Supplementary Fig. 8 | DFT calculation shows the dipole moment of the various electronic states of the complex 1(PF₆)₃. Direction of dipole moment for all the states are from Fe to Co centre (AF- antiferromagnetically coupled; FM- Ferromagnetically coupled).

Point 4. Directly linked to point 3 is the consideration that the observed room temperature value of χT is more in agreement with an $S=2$ with g slightly larger than 2 (as expected for Fe(II)) than for an $S=2$ arising from Fe(III) and radical species, for which $g=2$ is expected (in this case, the value of 3.12 and the decrease on decreasing temperature might suggest an incomplete AF coupling at this temperature). Here again DFT calculation could shed light on the expected Fe(III)-radical coupling.

Thank you for your comments. As mentioned above, DFT calculations cannot provide a reasonable description of the Fe (III)-radical electronic structure. Therefore, no efforts have been made to obtain the coupling constant of the antiferromagnetic coupling based on DFT calculations. The CASSCF/NEVPT2 approach provides a ferromagnetically coupled state lying at 774 cm^{-1} above the ground level, which partially contributes to the magnetic susceptibility at room temperature. Indeed, the quasi-degenerate perturbation treatment based on the CASSCF/NEVPT2 wavefunction renders a susceptibility-temperature product of $3.13\text{ cm}^3\text{ K mol}^{-1}$. (Experimentally obtained susceptibility-temperature product is determined exactly to be $3.24\text{ cm}^3\text{ K mol}^{-1}$ at 300 K. Previously, we used the value to $3.12\text{ cm}^3\text{ K mol}^{-1}$ which is the average χT product throughout the high temperature regime; therefore, we made a correction in the manuscript). However, the possibility of the contribution from the thermally induced population on the Fe(II)HS-DHBQ(2-) state cannot be ruled out because the g factor of Fe(II) ion is generally larger than 2 due to the unquenched orbital angular momentum contribution. We hope you will be satisfied with our explanation.

Point 5. An important point also concerns the definition of the supposed shift from Fe(III)HS-DHBQ(3-) species to Fe(II)HS-DHBQ(2-) configuration at high temperature: is there any peculiar reason why the authors refrain to identify this as a Valence tautomerism (intramolecular redox isomerism)? If there is a difference, this should be made explicit; I suppose that simple quantum mixing, such as the one described in ref. 39, should be temperature independent. I further notice that this electron hopping occurs, according to authors, at a rate faster than the Mössbauer time window ($\tau \sim 10^{-8}\text{ s}$) but slower than the IR time window ($\tau \sim 10^{-12}\text{ s}$). In this respect, it looks like EPR should be a very appropriate technique to monitor this process: author should consider to apply it to this system and, at any rate, comment on this possibility.

Thank you for your comment. In our [FeCo] system, determining electronic structure is quite complicated due to the significant overlap between metal and ligand electron density. This quantum mixing has turned this metal-ligand coordination moiety into a partially covalent system which refrained us to explicitly declare this system as a classical valence-tautomeric system. The spatial shift of electron density distribution could be a more appropriate description of the observed phenomenon. Therefore, we have mentioned the high-temperature behavior as the electron redistribution from the population change between the two limiting configurations $[\text{Fe}^{3+}_{\text{HS}}\text{-d}^{\text{3-}}\text{hbq}^{\text{3-}}\text{-Co}^{\text{3+}}_{\text{LS}}]$ and $[\text{Fe}^{2+}_{\text{HS}}\text{-d}^{\text{2-}}\text{hbq}^{\text{2-}}\text{-Co}^{\text{3+}}_{\text{LS}}]$ state that takes place over a large span of temperature.

Thank you for your suggestions to measure the EPR spectra. Indeed, EPR can be a technique with suitable timescale. Currently, we are collaborating with a specialized EPR group for further in-depth investigations on this complicated electron dynamics of [FeCo] system. We believe this part of results deserves to be published separately. We have included a comment on EPR as our future plan in the conclusion section as, “It should be noted that, considering its time scale, electron paramagnetic resonance should be a very appropriate technique to investigate the electronic dynamics in our compound.”

Point 6. Last, but not least: I found it quite astonishing not to find referenced the paper of Prof. A. Dei on the original dinuclear Co system (ACIE 2004) featuring CTH as ancillary ligand, which is far more appropriate than the current ref. 33, where TPA derivatives were used.

Thank you for pointing out this serious mistake we've done. We have modified this citation with your suggested one.

Reviewer #2

The approach to the development of a new material featuring electronic pyroelectricity based on switchable polarization is very elegant, since it involves a crystal engineering technique to crystallise a complex showing valence tautomerism in a polar space group. Anyway, I believe that the present manuscript does not fulfil the requirements to be published in the Nature Communications journal in terms of novelty of the results. First of all, a similar approach has been already reported previously by the same authors (reffer. 25 and 30 in the text). In ref. 30, in fact, the authors used the same enantiopure CTH ligands to crystallise the corresponding CrCo dinuclear complex in a polar space group, while in reference 25 the same authors reported the pyroelectricity associated with a valence tautomeric transition of a cobalt complex for the first time. These findings, in my opinion, affect the novelty of the presented results in a significant manner. Moreover, the novelty consisting in the continuous current release reported in the present work is somehow reduced by the fact that the pyroelectric coefficient of the compound $[\text{Co}(\text{phendiox})(\text{rac-cth})](\text{ClO}_4) \cdot 0.5\text{EtOH}$, reported in reference 25, stayed above the 5.0 nC/Kcm^{-2} limit for a wide temperature range as well (200 – 330 K). With these considerations in mind, I suggest publishing the present manuscript in a more specialised journal.

Thank you very much for reviewing our manuscript and your comments. Although we have used the same crystal engineering strategy to make polar [FeCo] system as mentioned in Ref. 30, the novelty of our work is not centered around the strategy of crystal engineering; rather, we focused on the exotic pyroelectric behavior of the system in terms of continuous current release over a large temperature range which originates from a novel interplay between three distinct electronic structures. In Ref. 25, pyroelectricity is introduced at molecular scale in terms of very common phenomenon of valence tautomerism (VT) in a cobalt-dioxolene complex; rather in our manuscript, we have introduced a highly covalent system which demonstrates how different electron dynamics plays crucial roles in the manifestation of polarization based pyroelectric current up to room temperature and above. One crucial point to be focused that the electron dynamics introduced here is not any typical SC or VT behavior, rather it portrays a systematic electron redistribution in terms of interconversion between three distinct electronic states, which made this system very unique with respect to others.

Notably, the high-temperature electronic transition is well hidden from magnetometry because the spin multiplicities of the two states involved are the same. It is the pyroelectric measurement that pushes us to notice the peculiar electronic dynamics that happens at high temperature. In this regard, the pyroelectricity plays a dual role in our study; on the one hand, it is the desired

property of our functional materials. On the other hand, it is also the characterization method for the peculiar transition behavior that cannot be easily established by routine characterizations.

Thinking from materials aspect, though there are many reports of the conventional pyroelectric effect in ferroelectric systems, but those suffer from a significant temperature-dependent decrease upon heating from the transition temperature (Curie temperature) where the ferroelectric phase turned to the paraelectric phase and spontaneous polarization vanishes. Here in this work, the peculiar electronic dynamics renders significant pyroelectricity of purely molecular origin after the magnetic transition temperature.

We hope our explanation regarding the novelty of our work would be satisfactory to you.

Review #3 (Answers)

‘In this manuscript, the authors synthesized a novel polar crystal [(Fe(RR-cth))(Co(SS-cth))(μ-dhbq)](PF₆)₃ and reported its pyroelectric effect. It is interesting that different from the conventional pyroelectric effect in ferroelectrics, the polarization of the molecular material [FeCo] dinuclear complex originates from a combination of Fe spin transition and redistribution of electron density between redox isomers of high-spin Fe through a charge transfer between the Fe atom and redox active ligand, which can be changed by the change of temperature, yielding the pyroelectric response. Even though the pyroelectric coefficient is inferior to those of conventional ferroelectrics including ceramics and single crystals, the sizable pyroelectric coefficient can be observed in a wide temperature range over a ~200 oC temperature span. The basic idea and hypothesis of the results are supported by applying several experimental data and DFT calculations. It is believed that this work would motivate further explorations of pyroelectric materials, especially pyroelectric features in molecular materials. Though the work is interesting, there are still several concerns that are suggested to be addressed’

Thank you very much for reviewing our manuscript. We believe your valuable suggestions would add more value to our work and increase the quality of this manuscript. We have tried to answer your concerns and have done suitable corrections according to your suggestions.

Point 1. The authors claimed that “To unambiguously determine the direction of the pyroelectric current with respect to the single crystal, a plate-like piece of complex 1(PF₆)₃ crystal was indexed with silver paste attached to one side of the crystal surface and carbon paste attached to the parallel surface and the faces (010) and (0-10) were distinguished” in the section of Pyroelectric property. How did the authors determine the crystal faces? Please provide credible characterization. Moreover, how did the authors get the plate-like piece of the complex crystal? A photo of single-crystal should be given, in which the crystal faces should be marked

At first, we determined the crystal faces of the block shape single-crystal of [FeCo] complex using a Rigaku FR-E+ diffractometer and CrysAlis Pro software package, and a figure (Supplementary Fig. 4) showing the face index of the crystalline piece has been included in the supplementary materials. Details are provided in the ‘Method’ section.

We obtained plate-like single crystals of the complex **1(PF₆)₃** during the recrystallization process from a mixed solvent of several drops of MeCN and hot H₂O/MeOH. We have mentioned in the synthesis part under ‘method’ section of the manuscript.

Supplementary Fig. 4 | The single-crystal of $1(\text{PF}_6)_3$ used in pyroelectric measurement (inset- crystal phases are marked on a block-shape single crystal).

Point 2. As shown in Fig. 4b, the pyroelectric current could be detected even over the transition temperature. It is well-known that the crystal defects seriously affect the pyroelectric current with the increasing temperature. Therefore, the measurements of dielectric constant/loss and resistivity versus temperature should be supplied.

Thank you very much for pointing out this aspect. We agree that the crystal defects or the presence of defect dipoles in ferroelectrics substantially affects the pyroelectric current with the increasing temperature. Therefore, to exclude the possibility that the continuous current release above the spin-transition stems from crystal defects, we measured AC pyroelectric property by continuous oscillation method from liquid nitrogen temperature to room temperature. This technique ensures accurate measurement of the pyroelectric current irrespective of any crystal defects. [Ref. 43-Garn L. E. and Sharp E. J. Use of low-frequency sinusoidal temperature waves to separate pyroelectric currents from nonpyroelectric currents. Part I. Theory, 53, 8974-8979, J. Appl. Phys. 1982, <https://doi.org/10.1063/1.330454>]. We described the features of the AC measurement on page 6~7 of the manuscript of the paper. Therefore, we preferred to measure AC pyroelectric response of our complex over dielectric measurement. Supplementary Fig. 5 shows the comparison between DC and AC pyroelectric responses which is a good match with our claim of continuous pyroelectricity over a wide temperature range.

Moreover, for strong evidence against the defect induced current, we have measured pyroelectricity in a [CoZn] complex. We couldn't find such continuous current release in this system, which supports our claim of this exotic pyroelectric property is the intrinsic property of [FeCo] system.

Regarding the resistivity measurement, we have measured the conductivity of the sample used in the pyroelectric measurement, which shows a very negligible conductance at low temperature as well as throughout the whole temperature range from 50 K to 300 K which is a signature of an insulator material. Indeed, different from band-like electronic structure in the correlated systems, the electronic density overlap between different molecular entities is almost negligible, which makes our system extremely insulate. We hope our explanation would be satisfactory to you.

Supplementary Fig. 5 | Qualitative comparison between pyroelectric responses of 1(PF₆)₃ measured via (a) continuous temperature ramping mode (DC mode) and (b) continuous oscillation mode (AC mode). (c) and (d) show the electric polarization obtained from the respective pyroelectric data. Specific heat of the sample used to estimate the polarization in (d) is considered to be constant with respect to temperature change.

Point 3. It is believed that the statement “the conventional pyroelectric effect in ferroelectric systems suffers from a significant temperature-dependent decrease upon cooling from the transition temperature, severely restricting their working temperature ranges” is inappropriate. In conventional ferroelectrics, for example, lead zirconate titanate (PZT) and lead magnesium niobite- lead titanate (PMN-PT), much larger pyroelectric coefficients can be obtained in a wide temperature range. The commercial PIR detectors equipped with PZT ceramics can work in both winter and summer regardless of regions from the equator to the high latitude area.

Thank you for pointing out the mistake. We have deleted this inappropriate statement from our manuscript.

Point 4. According to the demonstration in the manuscript, it is found that the polarization property of $[(\text{Fe}(\text{RR-cth}))(\text{Co}(\text{SS-cth}))(\mu\text{-dhibq})](\text{PF}_6)_3$ is quite different from that of the traditional pyroelectric materials. The polarization of $[(\text{Fe}(\text{RR-cth}))(\text{Co}(\text{SS-cth}))(\mu\text{-dhibq})](\text{PF}_6)_3$ crystal increases with the increase of temperature, which, however, decreases in ferroelectric materials (except part of antiferroelectric). Why does the polarization increase with the temperature in the molecular material? The nature of the positive pyroelectric coefficient is suggested to be discussed in detail.

In most traditional pyroelectric materials, pyroelectricity originates from the ferroelectric transition which requires the emergence of collective lattice modes (e.g. spontaneous molecular reorientation, ion-displacement or electron movements). Consequently, the polarization will decrease upon heating towards the transition temperature. However, in our molecular approach, we can tune molecular dipole moment by changing the temperature regime and it represents a molecular phenomenon. The polarization change direction is solely determined by the electronic structures that dominate at different temperature regimes. We have recently reported a couple of polarization switching behaviors in Co valence tautomeric compounds. The valence tautomeric transition, i.e. electron transfer coupled spin transition, is known to be driven by the spin and vibrational entropy, where low spin Co^{3+} state is in the low temperature phase and entropically favorable high spin Co^{2+} state is in the high temperature phase. In Ref. 25, we reported a cobalt mononuclear complex $[\text{Co}(\text{cth})(\text{phendiox})](\text{ClO}_4) \cdot 0.5 \text{ EtOH}$, where low spin Co^{3+} state, $\{\text{Co}^{3+}(\text{phendiox})^{2-}$ state $\}$, with larger dipole moment is in low temperature phase and entropically favorable high spin Co^{2+} state, $\{\text{Co}^{2+}(\text{phendiox})^-$ state $\}$, with smaller dipole moment is in high

temperature phase. Therefore, decrease in dipole moment is observed from LT to HT phase change. In Ref. 30, a [CrCo] dinuclear complex is reported, where low spin Co^{3+} state, $\{\text{Co}^{3+}\text{-(diox)}^3\text{-Cr}^{3+}$ state}, with smaller dipole moment is in low temperature phase and entropically favorable high spin Co^{2+} state, $\{\text{Co}^{2+}\text{-(diox)}^2\text{-Cr}^{3+}$ state}, with larger dipole moment is in high temperature phase. Therefore, in contrast to above mononuclear Co complex, increase in dipole moment is observed from LT to HT phase change, which is similar to this [FeCo] molecule where high temperature phase has larger polarization. Therefore, in the molecular systems, we have the choice of designability upon polarization switching. However, although the electronic structure of high- and low-temperature phases are discussed in text, we have not clearly mentioned the driving force of the transition, which is directly relevant to this question. Therefore, we included the sentence, “the spin transition is driven by an entropy originating from the change in spin and vibrational contribution”, and cite a reference (Ref. 37- P. Gülich, A. B. Gaspar and Y. Garcia, *Beilstein Journal of Organic Chemistry*, 9, 342-391, 2013).

Point 5. More experimental details should be given:

- 1) The information of the reagents used in the experiment should be added.
- 2) The silver and carbon pastes were used as the electrodes in the experiment. How did the authors control the sizes of the electrodes? And what is the size of the electrode?

Thank you very much for pointing out these corrections. We have added information of the reagents. Regarding, point 5.2, we have determined the size of the electrodes by the software equipped with the high-resolution optical microscope. We have added Supplementary Fig. 4 for a better understanding on the experimental setup where the size of the electrode is mentioned.

Point 6. The authors should pay more attention to the format of the units:

- 1) There are three kinds of unit formats in this manuscript: “5 K·min⁻¹”, “5 K/min”, and “1.15 $\mu\text{C cm}^{-2}$ ”.
- 2) In the pictures, what should be used to distinguish the physical quantities and corresponding units, “/” or “()” ?
- 3) “ml” should be “mL”. Please check the mistakes like this in the whole manuscript.

Unit format of “5 K·min⁻¹” is used throughout the manuscript. Corresponding units are revised and designated by ‘()’ in the whole manuscript. We have changed ‘ml’ to ‘mL’ and the revisions are highlighted by markers.

Reviewers' comments:

Reviewer #1 (Remarks to the Author):

In this revised version authors largely reorganized the manuscript, which is now much smoother in terms of readability. The sections are organized according to the different techniques used and for almost all of them the discussion is self-consistent, with one notable exception reported below. The modification of the title is also going in the direction of a more direct comprehension of the subject and of the message of the paper.

Authors tackled most of the points I raised in my previous report. However, it is now more evident that the theoretical discussion does not make a convincing case for explaining the observed behaviour at high temperature. In this respect, it is my opinion that in the present form this part is not at the level of a Nature Communications paper. The reasons are the following:

1- DFT is not able to provide an estimate of the AF coupling, in Fe(III)HS-DHBQ(3-), since it does not obtain this state as the ground state. The inability to obtain such state is not obvious to me, since calculations of exchange coupling constant even in highly delocalized systems containing semiquinone have been reported since many years. At any rate, this is a major flaw which affects IR calculations. Indeed, authors were forced to calculate the IR spectrum of the HT phase by considering the ferromagnetically coupled [Fe3+HS-dhbq3--Co3+LS] state. However, according to post HF calculations mentioned in their response letter, this state is lying far above the ground state in energy (775 cm⁻¹). It is evident that this poses some doubts as for the analysis of the IR spectra; further, post-HF results should be discussed, including those demonstrating that a partial Fe(II)HS character i.e. quantum mixing of states, at high temperature.

2- In general, theoretical results are still reported in an unclear manner. A separated paragraph should be provided stating more clearly what they did, and the assumption of localized vs delocalized description. In the present version, the description in terms of localized (ionic) states vs delocalized (covalent/quantum mixed) states are intermingled in the manuscript making it difficult to understand authors reasoning. As a simple example in this respect: I would refrain from indicating Fe3LS-dhbq3-Co3+Ls as "highly covalent", since authors are here using a localized description. The covalency characterizes the ground state because the electron delocalization toward the metal provides the GS with contribution from Fe(II)LS-DHBQ2-. Many comparable instances of this confusing description are found along the manuscript.

3- Even after having read the response letter and the new version, it is not possible to figure out how, if the ground state after SCO is still a quantum-mixed state (i.e. contains contribution from both "S=2" configurations), this should vary with temperature. It looks here as if authors were still attributing the electron redistribution to a population change between the two "limit" configurations. However, the ground state is unique, and is (according to authors) a linear combination of two configurations. In this description there is no way the relative weight can change with temperature. The only possibility is that electron redistribution on thermal excitation occurs between two different states with different weights of the two configurations. However, the presence of an excited "quantum-mixed" state is not commented upon by the authors. In this respect, it might well be that both the steady current and the slow increase in χT at high T, above SCO temperature, is simply due to the residual LS Fe which transform to HS. This would be a minor fraction, escaping Mossbauer and XAS detection; the results of IR, which are somehow biased by the problems outlined above DFT, are not that clear on my opinion.

Finally, my suggestion is that authors try to solve these issues and, at the same time, reorganize the material by dedicating a specific section to calculations. Alternatively, they should remove most of calculation results, leaving only the hypothesis as speculation based on previous literature results.

Reviewer #2 (Remarks to the Author):

In my previous evaluation of the manuscript by Prof. Shinji Kanegawa and Prof. Osamu Sato, I suggested the Authors to submit it to a more specialised journal, because two previous publications limited its innovation, in my opinion. My point of view, albeit justified in detail, is a personal opinion on the significance of the presented results, and, as such, has been overridden by the ones of Reviewers 1 and 3. I am fine with this decision, since the pyroelectricity of Valence Tautomeric complexes is a new and very promising topic in this field, and the results presented in the reviewed manuscript are clear and strongly supported by a thorough experimental and theoretical characterisation.

However, I disagree with the authors on the question of discovering a new chemical phenomenon of rearrangement of charge density triggered by temperature variation (the revised part included starting from line 255). I believe that the present system falls into the description of Valence Tautomerism, since it involves a charge redistribution between a metal ion and an organic ligand (even if it is partial, due to the quantum mixing of the $\text{Fe}^{3+}\text{HS} / \text{Fe}^{2+}\text{LS}$ configurations and spread on a wide temperature interval). Indeed, it is often stated in the literature that a low degree of covalence is required to get the VT behaviour (E. Evangelio, D. Ruiz-Molina, Valence tautomerism: More actors than just electroactive ligands and metal ions, *Comptes Rendus Chimie* 2008, 11, 1137-1154, <https://doi.org/10.1016/j.crci.2008.09.007>), but VT has been detected also in coordination compounds with a significant degree of covalence (H.-J. Himmel, Valence tautomerism in copper coordination chemistry, *Inorg. Chim. Acta* 2018, 481, 56-68, <http://dx.doi.org/10.1016/j.ica.2017.07.069>) and MIIICat / MIISQ state mixing (I. Ando, T. Fukuishi, K. Ujimoto, H. Kurihara, Oxidation states and redox behavior of ruthenium ammine complexes with redox-active dioxolene ligands, *Inorganica Chimica Acta* 390 (2012) 47-52, <http://dx.doi.org/10.1016/j.ica.2012.04.002>).

Moreover, the term Valence Tautomerism is used in Ref. 49 (DeGayner, J. A., Wang, K. & Harris, T. D. A Ferric Semiquinoid Single-Chain Magnet 521 via Thermally-Switchable Metal-Ligand Electron Transfer. *Journal of the American Chemical Society* 140, 6550-6553 (2018)) to address a temperature induced intramolecular electron transfer, along with a spin transition of the iron centre for an Fe(III)-semiquinonoid one-dimensional coordination polymer, featuring an anilate-type ligand, structurally related to the one used in the present work.

In my opinion, the mixing of the two redox states of the iron ion is responsible for the gradual nature of the temperature induced transition. I do not believe that the quantum mechanical mixing of the limiting $\text{Fe}^{3+}\text{HS} / \text{Fe}^{2+}\text{LS}$ configurations and the covalence of the metal-ligand bonds identifies a new physical phenomenon and takes the present molecular system outside the Valence Tautomerism class. As a consequence, I would like to ask the Authors the following:

1) line 73: replace "which is followed by a temperature-induced continuous population change between two redox isomers, with the limiting structure symbolically denoted as $[\text{Fe}^{3+}\text{HS}-\text{d}^{\text{h}}\text{b}^{\text{q}}\text{3}^{\text{-}}-\text{Co}^{3+}\text{LS}]$ and $[\text{Fe}^{2+}\text{HS}-\text{d}^{\text{h}}\text{b}^{\text{q}}\text{2}^{\text{-}}-\text{Co}^{3+}\text{LS}]$." with "which is followed by a temperature-induced continuous population change between two redox isomers, with the limiting structure symbolically denoted as $[\text{Fe}^{3+}\text{HS}-\text{d}^{\text{h}}\text{b}^{\text{q}}\text{3}^{\text{-}}-\text{Co}^{3+}\text{LS}]$ and $[\text{Fe}^{2+}\text{HS}-\text{d}^{\text{h}}\text{b}^{\text{q}}\text{2}^{\text{-}}-\text{Co}^{3+}\text{LS}]$, a phenomenon known as Valence Tautomerism."

2) line 252, remove: "Notably,".

3) line 255: remove the new section starting from "Though resembling the classical valence tautomeric process ..." until "The spatial shift of electron density distribution could be a more appropriate description of the observed phenomenon".

4) line 284: replace "the unusual transition" with "the Valence Tautomeric transition".

Minor suggestions:

- line 118: replace "Magnetic Property" with "Magnetic Properties"
- line 133: replace "Pyroelectric Property" with "Pyroelectric Properties"

Reviewer #3 (Remarks to the Author):

The authors have addressed my concerns

Response to Reviewers

Reviewer #1 (Remarks to the Author):

In this revised version authors largely reorganized the manuscript, which is now much smoother in terms of readability. The sections are organized according to the different techniques used and for almost all of them the discussion is self-consistent, with one notable exception reported below. The modification of the title is also going in the direction of a more direct comprehension of the subject and of the message of the paper. Authors tackled most of the points I raised in my previous report.

Response: We thank you very much for your well-considered review and further suggestions to improve our manuscript.

Point 1. However, it is now more evident that the theoretical discussion does not make a convincing case for explaining the observed behaviour at high temperature. In this respect, it is my opinion that in the present form this part is not at the level of a Nature Communications paper. The reasons are the following:

1- DFT is not able to provide an estimate of the AF coupling, in Fe(III)HS-DHBQ(3-), since it does not obtain this state as the ground state. The inability to obtain such state is not obvious to me, since calculations of exchange coupling constant even in highly delocalized systems containing semiquinone have been reported since many years. At any rate, this is a major flaw which affects IR calculations. Indeed, authors were forced to calculate the IR spectrum of the HT phase by considering the ferromagnetically coupled [Fe³⁺HS-dhbq³⁻-Co³⁺LS] state. However, according to post HF calculations mentioned in their response letter, this state is lying far above the ground state in energy (775 cm⁻¹). It is evident that this poses some doubts as for the analysis of the IR spectra; further, post-HF results should be discussed, including those demonstrating that a partial Fe(II)HS character i.e. quantum mixing of states, at high temperature.

Response: We thank the reviewer for their comments. As we mentioned earlier that the AF Fe(III)HS-DHBQ(3-) state cannot be obtained by using DFT methods due to the non-dynamical correlation error for strongly correlated systems. The conclusions based on the comparison between the experimental and calculated IR absorption could therefore be misleading. Conversely, even though the post-HF calculation can overcome such obstacles, its huge computational cost for frequency analysis of such a big molecule makes it a formidable task. Therefore, to lay our conclusion on a more solid ground, we have removed supp. Figs. 9 and 10 and also detailed calculation from our revised manuscript to avoid ambiguity in our conclusions. Instead, we draw

our conclusion based on the comparison of infrared absorption spectra and difference IR spectra between our [FeCo] system and the well-established valence tautomeric [CrCo] compound (Supplementary Figs. 9 and 10). It can be found that the HT behavior of the [FeCo] system clearly resembles that of the VT transition; the peaks around 1485 cm^{-1} (belongs to C-O vibrational mode of $\text{d}h\text{b}q^{3-}$) continuously decrease in intensity, whereas a new peak around 1556 cm^{-1} (belongs to C-O vibrational mode of $\text{d}h\text{b}q^{2-}$) emerges and increases upon heating. However, completely different features were found for the LT behavior of the [FeCo] system. It should be noted that the band at around 1540 cm^{-1} has been used as diagnostic of bridging $\text{d}h\text{b}q^{2-}$. Dei et al. reported “The

IR spectra of the three compounds are strictly similar to each other and in particular show a very intense band at 1540 cm^{-1} to be assigned to the carbonyl stretching mode of the bridging DHBQ^{2-} ligand”, where the three compounds they considered were $[\text{Mn}_2(\text{CTH})_2(\text{DHBQ})]^{2+}$, $[\text{Fe}_2(\text{CTH})_2(\text{DHBQ})]^{2+}$, and $[\text{Ni}_2(\text{CTH})_2(\text{DHBQ})]^{2+}$ (A. Dei et al., *Inorg. Chem.*, 30, 2589-2594 (1991)). Even we confirmed the presence of strong band around 1540 cm^{-1} (responsible for DHBQ^{2-}) in our reference complexes $[\text{Co}^{\text{III}}\text{-dhbq}^{2-}\text{-Zn}^{\text{II}}](\text{PF}_6)_3$ and $[\text{Fe}^{\text{II}}\text{-dhbq}^{2-}\text{-Co}^{\text{II}}](\text{PF}_6)_2$ (Supplementary Fig. 11).

Consequently, we concluded that the electron dynamics dominating the HT and LT behaviors are different, and electron-transfer process dominates the HT behavior whereas the SCO dominates the LT behavior. Therefore, the Reviewer's suggestion mentioned in **Point 3**, "both the steady current and the slow increase in χT at high T, above SCO temperature, is simply due to the residual LS Fe which transform to HS", is not valid.

Furthermore, it is well known that quadrupole splitting in Mössbauer spectra usually decreases in width or almost constant with increasing temperature for Fe complexes. However, at HT phase above spin transition, the quadrupole splitting clearly increases with increasing temperature, which suggests an increase in the contribution of $\text{Fe}^{2+}_{\text{HS}}$ with its larger quadrupole splitting value. This increase cannot be explained by the transformation of residual LS Fe to HS.

Supplementary Fig 11 | Comparison of IR spectra (recorded at 300 K) between reference complexes $[\text{Co}^{\text{III}}\text{-d}(\text{h}(\text{b}(\text{q})^2\text{-Zn}^{\text{II}})](\text{PF}_6)_3$ and $[\text{Fe}^{\text{II}}\text{-d}(\text{h}(\text{b}(\text{q})^2\text{-Co}^{\text{II}})](\text{PF}_6)_2$ indicating peak position at around 1540 cm^{-1} characteristic of bridging $[\text{d}(\text{h}(\text{b}(\text{q})^2\text{-}$ ligand.

Point 2. In general, theoretical results are still reported in an unclear manner. A separated paragraph should be provided stating more clearly what they did, and the assumption of localized

vs delocalized description. In the present version, the description in terms of localized (ionic) states vs delocalized (covalent/quantum mixed) states are intermingled in the manuscript making it difficult to understand authors reasoning. As a simple example in this respect: I would refrain from indicating Fe₃LS-dhbq₃-Co³⁺LS as “highly covalent”, since authors are here using a localized description. The covalency characterizes the ground state because the electron delocalization toward the metal provides the GS with contribution from Fe(II)LS-DHBQ²⁻. Many comparable instances of this confusing description are found along the manuscript.

Response: We thank the reviewer for their suggestions. As mentioned in Response to **Point 1**, we have removed the calculation results due to the inability to obtain the AF Fe(III)HS-DHBQ(3-) state. Regarding the “confusing description”, we have revised the description and notation in the manuscript. In the introduction, we used the limiting configurations, [Fe³⁺_{LS}-dhbq³⁻-Co³⁺_{LS}], [Fe³⁺_{HS}-dhbq³⁻-Co³⁺_{LS}] and [Fe²⁺_{HS}-dhbq²⁻-Co³⁺_{LS}], to denote their electronic structures following convention, and an extra superscript C (standing for covalent) is used to remind the readers of the presence of strong covalency in such states. In the same manner, we gave notations ^C[Fe³⁺_{HS}-dhbq³⁻-Co³⁺_{LS}] and ^C[Fe²⁺_{HS}-dhbq²⁻-Co³⁺_{LS}] for the quantum mixing state at HT phase and an excited state at HT phase that is populated with increasing temperature, respectively. We used these notations throughout the manuscript to avoid confusion. We hope the revisions make the manuscript easier to follow.

Point 3. Even after having read the response letter and the new version, it is not possible to figure out how, if the ground state after SCO is still a quantum-mixed state (i.e. contains contribution from both “S=2” configurations), this should vary with temperature. It looks here as if authors were still attributing the electron redistribution to a population change between the two “limit” configurations. However, the ground state is unique, and is (according to authors) a linear combination of two configurations. In this description there is no way the relative weight can change with temperature. The only possibility is that electron redistribution on thermal excitation occurs between two different states with different weights of the two configurations. However, the presence of an excited “quantum-mixed” state is not commented upon by the authors. In this respect, it might well be that both the steady current and the slow increase in χ_iT at high T, above SCO temperature, is simply due to the residual LS Fe which transform to HS. This would be a minor faction, escaping Mossbauer and XAS detection; the results of IR, which are somehow biased by the problems outlined above DFT, are not that clear on my opinion.

Finally, my suggestion is that authors try to solve these issues and, at the same time, reorganize the material by dedicating a specific section to calculations. Alternatively, they should remove

most of calculation results, leaving only the hypothesis as speculation based on previous literature results.

Response: We thank the reviewer for their suggestions. We agree with the statement that electron redistribution or VT on thermal excitation occurs between two different states. Actually, this was the case we intended to convey in our original and previous revised versions. We have included the following statement about the presence of an excited “quantum-mixed” state or covalent state in the text to ensure that the intended meaning is understood:

Such features demonstrate that there is a thermally accessible excited state, ${}^C[\text{Fe}^{2+}_{\text{HS}}\text{-d}^2\text{hbq}^{2-}\text{-Co}^{3+}_{\text{LS}}]$, at HT phase. They also show that the ${}^C[\text{Fe}^{2+}_{\text{HS}}\text{-d}^2\text{hbq}^{2-}\text{-Co}^{3+}_{\text{LS}}]$ state gradually becomes the dominating phase in thermal equilibrium with the ${}^C[\text{Fe}^{3+}_{\text{HS}}\text{-d}^3\text{hbq}^{3-}\text{-Co}^{3+}_{\text{LS}}]$ state at the HT phase which is a signature of valence tautomerism that takes place over a wide temperature range.

Regarding the origin of steady current and the slow increase in $\chi_m T$ at high T above SCO temperature, as stated in response to **Point 1**, IR and Mössbauer results reveal that the population of ${}^C[\text{Fe}^{2+}_{\text{HS}}\text{-d}^2\text{hbq}^{2-}\text{-Co}^{3+}_{\text{LS}}]$ state should be the origin. Furthermore, we think quantitative analysis of the experimentally obtained polarization change can shed further light on this. Integration of the pyroelectric current in the temperature change from 120 to 300 K yields the polarization change of ca. $0.67 \mu\text{C cm}^{-2}$, which is significantly larger than that during the SCO process ($0.48 \mu\text{C cm}^{-2}$) [Ref. Figure A]. Considering the relatively smaller change in Fe-ligand bond lengths observed in this temperature domain, the origin of the pyroelectric effect should be assigned to a process with more electron-transfer character rather than the incomplete SCO behavior. In this regard, this assignment is also consistent with the gradual increase of the $\chi_m T$ value in high temperature domain, which is caused by the gradual increase in the population of $\text{Fe}^{2+}_{\text{HS}}$ with unquenched orbital angular momentum. We hope you will find our explanation satisfactory.

Figure A. Correlation of magnetic properties and polarization change for [FeCo] complex.

Reviewer #2 (Remarks to the Author):

In my previous evaluation of the manuscript by Prof. Shinji Kanegawa and Prof. Osamu Sato, I suggested the Authors to submit it to a more specialised journal, because two previous publications limited its innovation, in my opinion. My point of view, albeit justified in detail, is a personal opinion on the significance of the presented results, and, as such, has been overridden by the ones of Reviewers 1 and 3. I am fine with this decision, since the pyroelectricity of Valence Tautomeric complexes is a new and very promising topic in this field, and the results presented in the reviewed manuscript are clear and strongly supported by a thorough experimental and theoretical characterisation.

However, I disagree with the authors on the question of discovering a new chemical phenomenon of rearrangement of charge density triggered by temperature variation (the revised part included starting from line 255). I believe that the present system falls into the description of Valence Tautomerism, since it involves a charge redistribution between a metal ion and an organic ligand (even if it is partial, due to the quantum mixing of the $\text{Fe}^{3+}\text{HS} / \text{Fe}^{2+}\text{LS}$ configurations and spread on a wide temperature interval). Indeed, it is often stated in the literature that a low degree of covalence is required to get the VT behaviour (E. Evangelio, D. Ruiz-Molina, Valence tautomerism: More actors than just electroactive ligands and metal ions, *Comptes Rendus Chimie* 2008, 11, 1137–1154, <https://doi.org/10.1016/j.crci.2008.09.007>), but VT has been detected also in coordination compounds with a significant degree of covalence (H.-J. Himmel, Valence tautomerism in copper coordination chemistry, *Inorg. Chim. Acta* 2018, 481, 56–68, <http://dx.doi.org/10.1016/j.ica.2017.07.069>) and MIIICat / MIISQ state mixing (I. Ando, T. Fukuishi, K. Ujimoto, H. Kurihara, Oxidation states and redox behavior of ruthenium ammine complexes with redox-active dioxolene ligands, *Inorganica Chimica Acta* 390 (2012) 47–52, <http://dx.doi.org/10.1016/j.ica.2012.04.002>).

Moreover, the term Valence Tautomerism is used in Ref. 49 (DeGayner, J. A., Wang, K. & Harris, T. D. A Ferric Semiquinoid Single-Chain Magnet 521 via Thermally-Switchable Metal–Ligand Electron Transfer. *Journal of the American Chemical Society* 140, 6550–6553 (2018)) to address a temperature induced intramolecular electron transfer, along with a spin transition of the iron centre for an Fe(III)-semiquinonoid one-dimensional coordination polymer, featuring an anilate-type ligand, structurally related to the one used in the present work. In my opinion, the mixing of the two redox states of the iron ion is responsible for the gradual nature of the temperature induced transition. I do not believe that the quantum mechanical mixing of the limiting $\text{Fe}^{3+}\text{HS} / \text{Fe}^{2+}\text{LS}$ configurations and the covalence of the metal-ligand bonds identifies a new physical phenomenon and takes the present molecular system outside the Valence Tautomerism class.

Response: We thank you very much for your kind response and suggestions regarding some of the key points in the manuscript. In our previous response, we adopted the concept of ‘spatial shift of electron density distribution’, responsible for the high-temperature electron dynamics. As most

of the conventional VT systems have either negligible or quite low degrees of covalency, which refrained us to claim the system as a classical VT. Conversely, as your reference article mentioned, several systems (including blue-copper protein) exhibits significant σ - or π -bond contributions towards metal-ligand covalency and these systems are allegedly categorized under the family of VT. Moreover, quantum mixing in [FeCo] system is nothing mysterious but rather is equivalent to the strong covalent metal-ligand interaction observed in some Ru-amine complexes or bio-inorganic systems like oxyhemoglobin.

We also agree that the ferric semiquinoid single-chain magnet reported by *Harris et al.* is structurally related to our system as the HT behaviour of $\text{Fe}^{\text{III}}_{\text{HS}}$ to $\text{Fe}^{\text{II}}_{\text{HS}}$ transition upon temperature rise is accompanied by only a small distortion of the Fe coordination sphere as depicted in Supp. Fig. 2 for [FeCo] complex.

After considering the above-mentioned key points, we have made modifications to the explanation of the HT pyroelectric behavior and assigned it to a process of VT in terms of electron transfer which takes place over a wide temperature range. Our conclusions in the revised manuscript are based on the fact that the novel pyroelectric behavior of [FeCo] comprises of two different electron dynamics where the high temperature transition behavior comes from valence tautomerism and the low-temperature behavior is dominated by spin transition. Also we considered to carefully illustrate the significant quantum mixing at LT and HT phase as indicated by Mössbauer spectroscopy and more sophisticated HERFD-XANES measurements.

As a consequence, I would like to ask the Authors the following:

1) line 73: replace “which is followed by a temperature-induced continuous population change between two redox isomers, with the limiting structure symbolically denoted as $[\text{Fe}^{\text{3+}}\text{HS-dhbq}^{\text{3-}}\text{-Co}^{\text{3+}}\text{LS}]$ and $[\text{Fe}^{\text{2+}}\text{HS-dhbq}^{\text{2-}}\text{-Co}^{\text{3+}}\text{LS}]$.” with “which is followed by a temperature-induced continuous population change between two redox isomers, with the limiting structure symbolically denoted as $[\text{Fe}^{\text{3+}}\text{HS-dhbq}^{\text{3-}}\text{-Co}^{\text{3+}}\text{LS}]$ and $[\text{Fe}^{\text{2+}}\text{HS-dhbq}^{\text{2-}}\text{-Co}^{\text{3+}}\text{LS}]$, a phenomenon known as Valence Tautomerism.”

2) line 252, remove: “Notably,”.

3) line 255: remove the new section starting from “Though resembling the classical valence tautomeric process ...” until “The spatial shift of electron density distribution could be a more appropriate description of the observed phenomenon”.

4) line 284: replace “the unusual transition” with “the Valence Tautomeric transition”.

Minor suggestions:

- line 118: replace “Magnetic Property” with “Magnetic Properties”

- line 133: replace “Pyroelectric Property” with “Pyroelectric Properties”

Response: Thank you for suggesting these corrections. We have revised the manuscript accordingly.

Reviewer #3 (Remarks to the Author):

The authors have addressed my concerns.

Response: Thank you very much for your review.

REVIEWERS' COMMENTS

Reviewer #1 (Remarks to the Author):

In this third version authors took into account all my previous remarks, in particular by removing the somehow confusing computational section and by describing the high temperature transition responsible for the observed pyroelectric current as a gradual valence tautomerism. This helps much in following their reasoning, all the experimental evidences being in strong support of the interpretation of their data. It is my feeling that now both the results and their interpretation now deserves publication in Nature Communications. I have only a few minor suggestions to further improve the quality of the presentation:

l. 79-84: "Although... states": I suggest to move this sentence in a note rather than keeping it in the main text, to improve readability while maintaining clarity.

l.209-212: I suggest to rephrase as follows, for the sake of simplicity: "Such features reveal that in the HT phase there is a thermally accessible excited state, $C[Fe^{2+}HS-dhbq^{2-}-Co^{3+}LS]$, which gradually becomes the dominating phase in thermal equilibrium with the $C[Fe^{3+}HS-dhbq^{3-}-Co^{3+}LS]$ state. This a signature of valence tautomerism that takes place over a large span of temperature..."

l.217: I suggest to add, for completeness: "thus with an expected χ_T large than that of strongly AF coupled $Fe^{3+}HS-dhbq^{3-}$ "

Reviewer #2 (Remarks to the Author):

After the revisions made by the authors, who addressed all the points of my concern, I believe that the manuscript is worth of being published in the Nature Communications journal in its present form.

Response to Reviewers

Reviewer #1 (Remarks to the Author):

In this third version authors took into account all my previous remarks, in particular by removing the somehow confusing computational section and by describing the high temperature transition responsible for the observed pyroelectric current as a gradual valence tautomerism. This helps much in following their reasoning, all the experimental evidences being in strong support of the interpretation of their data. It is my feeling that now both the results and their interpretation now deserves publication in Nature Communications. I have only a few minor suggestions to further improve the quality of the presentation:

Response: We thank you very much for your well-considered review and further suggestions to improve our manuscript.

Point 1. 79-84: "Although... states": I suggest to move this sentence in a note rather than keeping it in the main text, to improve readability while maintaining clarity.

Response: Thank you for your comment. We have relocated this sentence in Supplementary note 1 to improve readability.

Point 2. 209-212: I suggest to rephrase as follows, for the sake of simplicity: "Such features reveal that in the HT phase there is a thermally accessible excited state, $C[Fe^{2+}HS-dhbq_2-Co_3+LS]$, which gradually becomes the dominating phase in thermal equilibrium with the $C[Fe^{3+}HS-dhbq_3-Co_3+LS]$ state. This a signature of valence tautomerism that takes place over a large span of temperature..."

Response: Thank you for your comment. We have rephrased the sentence according to your suggestion.

Point 3. I suggest to add, for completeness: "thus with an expected χ_T large than that of strongly AF coupled $Fe^{3+}HS-dhbq_3$ "

Response: Thank you for your comment. We have added this phrase for completeness.

Reviewer #2 (Remarks to the Author):

After the revisions made by the authors, who addressed all the points of my concern, I believe that the manuscript is worth of being published in the Nature Communications journal in its present form.

Response: We thank you very much for your well-considered review.